# Nucleotide proofreading functions by nematode RAD51 paralogs facilitate optimal RAD51 filament function

Mário Špírek[1,2,5], Martin R. G. Taylor[3,5], Ondrej Belan[3,4], Simon J. Boulton [3] & Lumir Krejci [1,2,4 ✉]

The RAD51 recombinase assembles as helical nucleoprotein filaments on single-stranded DNA (ssDNA) and mediates invasion and strand exchange with homologous duplex DNA (dsDNA) during homologous recombination (HR), as well as protection and restart of stalled replication forks. Strand invasion by RAD51-ssDNA complexes depends on ATP binding. However, RAD51 can bind ssDNA in non-productive ADP-bound or nucleotide-free states, and ATP-RAD51-ssDNA complexes hydrolyse ATP over time. Here, we define unappreciated mechanisms by which the RAD51 paralog complex RFS-1/RIP-1 limits the accumulation of RAD-51-ssDNA complexes with unfavorable nucleotide content. We find RAD51 paralogs promote the turnover of ADP-bound RAD-51 from ssDNA, in striking contrast to their ability to stabilize productive ATP-bound RAD-51 nucleoprotein filaments. In addition, RFS-1/RIP-1 inhibits binding of nucleotide-free RAD-51 to ssDNA. We propose that 'nucleotide proof-reading' activities of RAD51 paralogs co-operate to ensure the enrichment of active, ATP-bound RAD-51 filaments on ssDNA to promote HR.

[1] International Clinical Research Center, St. Anne's University Hospital in Brno, 62500 Brno, Czech Republic. [2] Department of Biology Masaryk University, 62500 Brno, Czech Republic. [3] The Francis Crick Institute, 1 Midland Road, London NW1 1AT, UK. [4] National Centre for Biomolecular Research, Masaryk University, 62500 Brno, Czech Republic. [5]These authors contributed equally: Mário Špírek, Martin R. G. Taylor. ✉email: lkrejci@chemi.muni.cz

During HR, RAD51-ssDNA filaments undergo invasion of homologous dsDNA, which provides a substrate for repair DNA synthesis[1]. Optimal RAD51-ssDNA filament function depends on regulatory proteins, including BRCA2 and the RAD51 paralogs in metazoans, to facilitate its accumulation and maintenance on ssDNA. The initial loading of RAD51 onto ssDNA requires exchange with the high-affinity ssDNA-binding protein RPA, a function fulfilled by human BRCA2[2–4] and yeast Rad52[5]. Once loaded onto ssDNA, regulatory proteins also serve to limit RAD51 dissociation and increase the lifetime of the presynaptic complex, including RAD51 paralogs.

RAD51 paralogs are a family of proteins with sequence or structural homology with RAD51 but lacking recombinase activity. We previously identified and characterized a RAD51 paralog complex from nematodes, RFS-1/RIP-1, discovering it has a critical role in stabilizing RAD-51-ssDNA filaments by reducing RAD-51 dissociation rate. Mechanistically, this activity is mediated by engagement of RFS-1/RIP-1 with the 5′ end of ATP-bound RAD-51-ssDNA filaments, which propagates the filament stabilizing effect with 5′–3′ polarity at least 40 nucleotides from the site of binding. In addition, RFS-1/RIP-1 also alters the biophysical properties of the RAD-51-ssDNA filament, remodeling it to a more 'open' and flexible conformation, which we proposed facilitates strand invasion[6,7].

RAD51 is active as a recombinase in the presence of the nucleotide co-factor ATP or non-hydrolysable analogues[8]. In contrast, although RAD51 is still competent to bind ssDNA in the absence of nucleotides or in the presence of ADP, such RAD51-ssDNA complexes from human, yeast and nematode are inactive for strand invasion activity[6,8–10]. In addition, RAD51 slowly hydrolyzes ATP in the presence of DNA[10,11], meaning the presynaptic complex is likely to have a finite lifetime as an active recombinase, becoming non-functional if the ratio of ATP:ADP falls below the threshold required for strand invasion activity.

Moreover, besides its function in the repair of DSBs, RAD51 has important roles at stalled replication forks where it promotes replication fork regression[12], protects tracts of newly synthesized DNA from degradation[13,14] and is required for restart of stalled or collapsed replication forks[15]. Recently it was shown that the enzymatic recombinogenic activity of RAD51 is not required to protect stalled forks[16], but expression of either ATP binding or hydrolysis-deficient RAD51 mutants causes fork stalling and increases gross chromosomal rearrangements[17]. However, it is still not clear precisely how ATP hydrolysis or nucleotide binding affects RAD51's role in replication.

Collectively, these observations emphasize the significance of studying RAD51's biochemical activities in different nucleotide-bound states, which may confer distinct functions. We hypothesized that biochemical regulatory mechanisms may exist to 'proofread' the nucleotide co-factor status of RAD51 before and/or after ssDNA binding, to specifically facilitate the accumulation of ATP-bound, but not ADP-bound or nucleotide-free, RAD51 on ssDNA. Here, we show that the nematode RFS-1/RIP-1 complex fulfils these functions by diverse mechanisms, in addition to its roles in remodeling and stabilizing ATP-bound RAD-51-ssDNA filaments. Our data reveal previously unappreciated nucleotide proofreading functions for RAD51 paralogs, suggesting that their ability to regulate HR is complex and multifaceted.

## Results
### RFS-1/RIP-1 stimulates RAD-51 binding to ssDNA in the presence of nucleotides.
In our previous studies on RFS-1/RIP-1, we used stopped-flow techniques and 5′-Cy3 fluorescently-labelled oligonucleotides to study the influence of RFS-1/RIP-1 on RAD-51-ssDNA filaments in real-time in the presence of ATP.

Upon the rapid mixing of RAD-51 and RFS-1/RIP-1 with 5′-Cy3 labelled 43mer ssDNA in the stopped-flow machine, we observed a two phase Cy3 fluorescence response: (1) an initial very rapid fluorescence increase phase, due to RAD-51 binding to the ssDNA, followed by (2) a second slower fluorescence reduction phase, representing RFS-1/RIP-1 association with the 5′ filament end[6,7]. These experiments were always performed in the presence of saturating RAD-51 concentrations (1000 nM) to ensure complete filament formation by RAD-51. This increased the dynamic range for observing filament binding by RFS-1/RIP-1 in the second phase. However, we noticed that RFS-1/RIP-1 also weakly but consistently stimulated the first RAD-51 ssDNA-binding phase under these conditions[6].

We reasoned that such a stimulatory effect of RFS-1/RIP-1 may be more robustly detected at limiting concentrations of RAD-51. We thus pre-incubated 250 nM RAD-51 (fourfold less than typically used in our prior study) with ATP in the presence of two different concentrations (10 and 50 nM) of RFS-1/RIP-1. We then rapidly mixed the proteins with 5′, 3′ or internally Cy3-labelled ssDNA using a stopped-flow instrument and monitored changes in fluorescence. On all three oligonucleotides, RFS-1/RIP-1 stimulated the magnitude of fluorescence increase due to RAD-51 binding (Fig. 1a-d), indicating that RFS-1/RIP-1 stimulates DNA binding by RAD-51. Importantly, our previous studies showed that for the latter two constructs, the Cy3 label is sufficiently distant from the 5′ end that its fluorescence is not modulated by filament binding of RFS-1/RIP-1[7]. Hence, any fluorescence modulation on these constructs reflects only the regulation of RAD-51 binding to the ssDNA. We have also previously shown that RFS-1/RIP-1 alone does not influence Cy3 fluorescence nor bind DNA under these conditions[6]. Hence RFS-1/RIP-1 stimulates DNA binding by RAD-51 on naked ssDNA, similarly to previously reported for human BRCA2[3].

To test whether the observed stimulatory activity on RAD-51 binding to ssDNA is ATP-dependent, identical experiments as described above were performed in the presence of ADP. Again, a clear RFS-1/RIP-1-mediated stimulation of RAD-51 binding to ssDNA for all three oligonucleotides was detected (Fig. 1e-h). We conclude that RFS-1/RIP-1 stimulates RAD-51 binding to ssDNA in the presence of either ATP or ADP nucleotide co-factors. It should be noted that some minor differences in relative fluorescence changes induced by RFS-1/RIP-1 on the different positions in the presence of both ATP and ADP (Fig. 1d, h) was also observed, which could possibly reflect polar effects of RFS-1/RIP-1 on RAD-51 filament formation. However, these were subtle and not clearly concentration-dependent.

### RFS-1/RIP-1 limits the binding of RAD-51 to ssDNA in the absence of nucleotide cofactors.
We next examined the effect of RFS-1/RIP-1 on RAD-51-ssDNA complexes in the absence of nucleotides. Previous electrophoretic mobility shift experiments in the absence of nucleotide cofactor showed more free ssDNA is observed when RFS-1/RIP-1 and RAD-51 are mixed compared to RAD-51 alone[6]. However, since this assay reveals the protein-ssDNA complexes at equilibrium and after crosslinking, it does not inform on whether RFS-1/RIP-1 limits RAD-51 binding, promotes RAD-51 dissociation, or both. Therefore, we first assessed the impact of RFS-1/RIP-1 on RAD-51 binding to ssDNA in the absence of nucleotide (Fig. 2a). In striking contrast to its stimulatory effect in the presence of ATP or ADP (Fig. 1), in the absence of nucleotide the co-incubation of RAD-51 with RFS-1/RIP-1 prior to mixing with Cy3-labelled ssDNA in stopped-flow experiments led to a dramatic and concentration-dependent reduction in the magnitude of fluorescence increase attributed to RAD-51 ssDNA binding (Fig. 2a, b). This suggests RFS-1/RIP-1

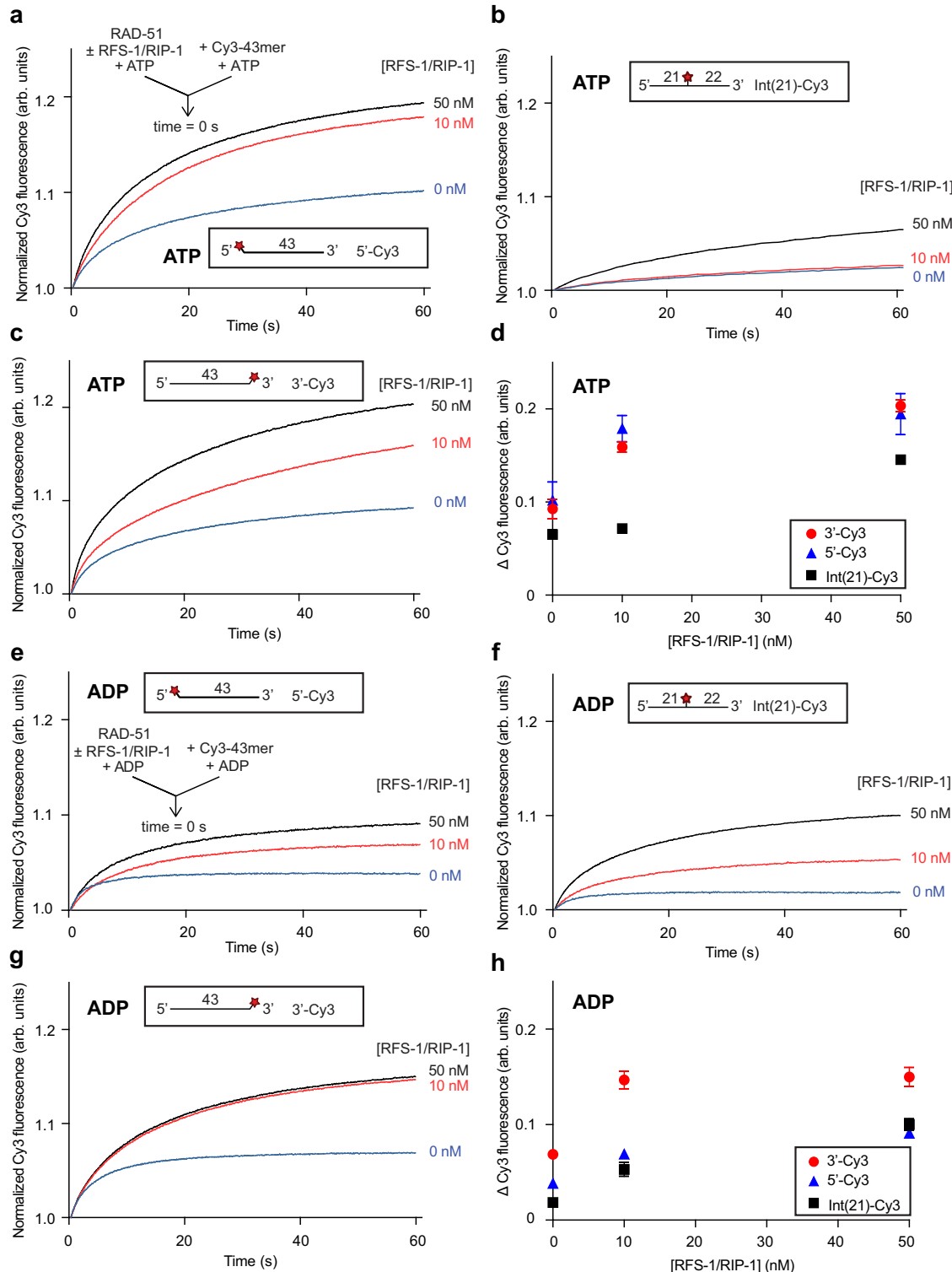

**Fig. 1 RFS-1/RIP-1 stimulates formation of RAD-51-ssDNA filaments in the presence of ATP or ADP.** Average normalized Cy3-43mer fluorescence profiles plotted as a function of time. The arrow indicates the components of the two syringes rapidly mixed at the 0 s time point in a stopped flow instrument. In all experiments RAD-51 (250 nM) alone or in the presence of RFS-1/RIP-1 (10 and 50 nM) was mixed with Cy3-43mer ssDNA (15 nM). **a** and **e** 5'-Cy3, (**b** and **f**) Int(21)-Cy3 or (**c** and **g**) 3'-Cy3 labelled substrates were used (**a**: $n = 5$, **b**: $n = 4$-5, **c**: $n = 4$-5, **e**: $n = 4$, **f**: $n = 4$-8, **g**: $n = 5$-6). Schematics of these different Cy3 label positions are shown inset. **d** and **h** Graphs of RFS-1/RIP-1 concentration-dependence of average Δ Cy3 fluorescence for the data presented in (**a**-**c**) and (**e**-**g**) respectively (mean; errors: s.d.). Experiments in (**a**-**d**) were performed in the presence of ATP, whereas (**e**-**h**) were performed with ADP. Source data are provided as a Source Data file.

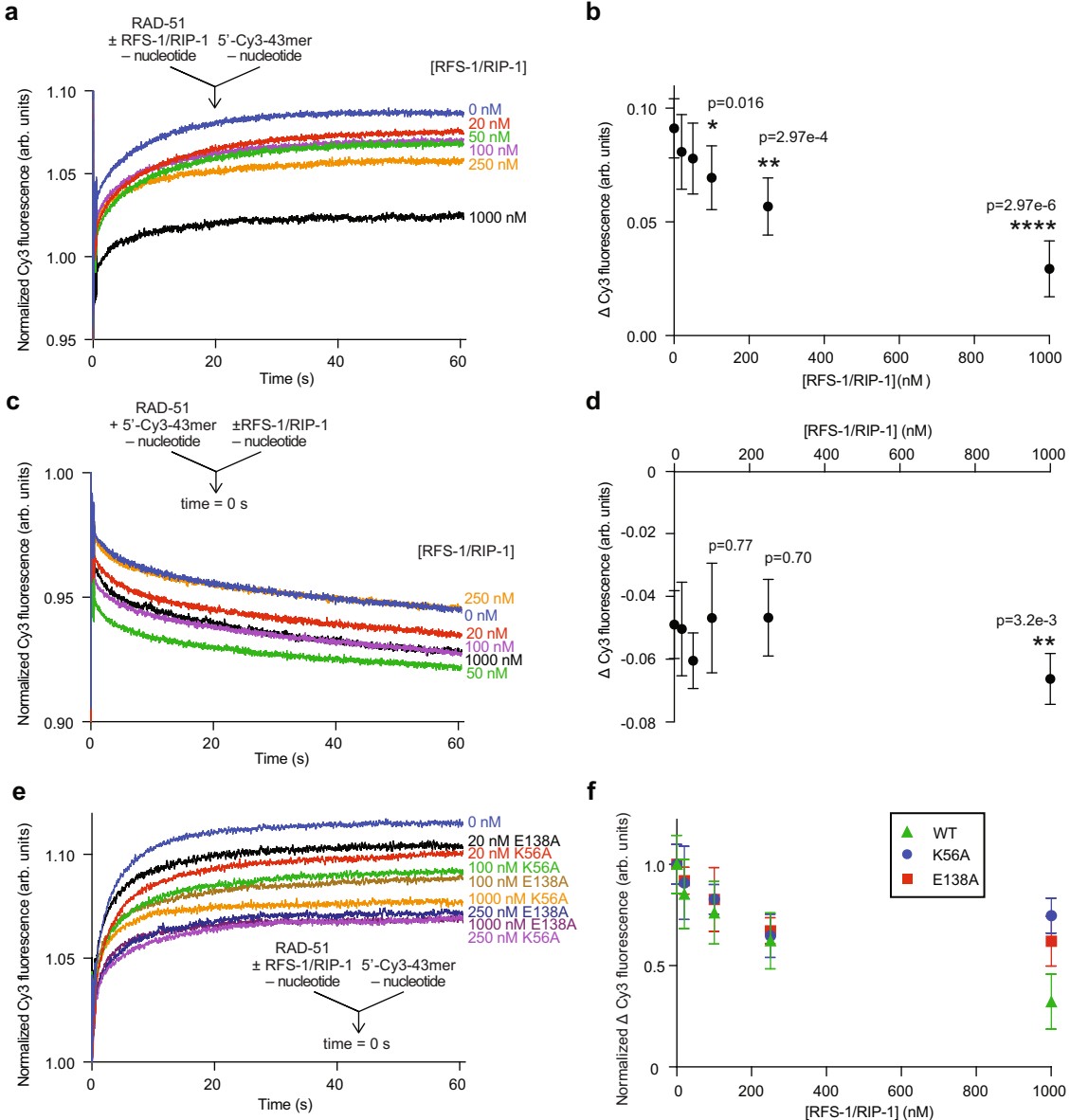

**Fig. 2 RFS-1/RIP-1 inhibits the binding of RAD-51 to ssDNA in the absence of nucleotide cofactors.** Average normalized Cy3-43mer fluorescence profiles plotted as a function of time. The arrow indicates the components of the two syringes rapidly mixed at the 0 s time point in a stopped flow instrument. RAD-51 (1 μM) was either pre-mixed with increasing amount of RFS-1/RIP-1 and then was mixed with 15 nM Cy3-43mer ssDNA (n = 6–7) (**a**) or RAD-51 was pre-mixed with ssDNA and afterwards mixed with RFS-1/RIP-1 in the absence of nucleotide and analyzed in stopped flow machine (n = 7–8) (**c**). Corresponding evaluations of average Δ Cy3 fluorescence as a function of concentration of RFS-1/RIP-1 for the experiments in (**a** and **c**) are presented in (**b** and **d**) respectively (mean; errors: s.d.). p values are relative to RAD-51 alone data obtained by Student's t test (two-tailed): *p < 0.05; **p < 0.01; ****p < 0.0001. **e** Effect of Walker box mutants of RFS-1 (K56A and E138A) on RAD-51 in the absence of nucleotide. The experiment was done similarly as in (**a**) (n = 5–8). **f** Normalized Δ Cy3 fluorescence relative to RAD-51 alone at different RFS-1/RIP-1 mutant concentrations from data shown in (**e**). Data for wild-type RAD-51 are taken from (**b**) for comparison (mean, errors: s.d.). Source data are provided as a Source Data file.

inhibits RAD-51 binding to ssDNA in the absence of nucleotide. Specifically, we observed a threefold reduction in fluorescent change at 1 μM RFS-1/RIP-1, with the half-change of this maximal amplitude occurring at an RFS-1/RIP-1 concentration of 0.2 μM, indicating sub-stoichiometric amounts of RAD51 paralog are required for this proofreading activity.

Next, we monitored the dissociation of pre-formed nucleotide-free RAD-51-ssDNA complexes upon mixing with buffer or different concentrations of RFS-1/RIP-1 in stopped-flow. Notably, RAD-51-ssDNA complexes formed without nucleotide were unstable and underwent spontaneous dissociation upon dilution with buffer (Fig. 2c, blue trace). Mixing with RFS-1/RIP-1 did not

further exacerbate this effect and the extent of fluorescence reduction was independent of RFS-1/RIP-1 concentration (Fig. 2c, d). We conclude that RFS-1/RIP-1 limits nucleotide-free RAD-51 accumulating on ssDNA by inhibiting its binding, rather than driving protein-ssDNA complex turnover.

We also studied the dependence of this activity on the RFS-1 Walker A (K56A) and B (E138A) boxes, which are critical for RFS-1/RIP-1's ability to stimulate RAD-51 in DNA strand exchange[6]. Both K56A and E138A mutants inhibited RAD-51 binding to ssDNA in the absence of ATP in a concentration-dependent manner (Fig. 2e, f) but to a reduced extent relative to the wild-type RFS-1/RIP-1 complex (Fig. 2f). We validated this

observation by electrophoretic mobility shift assays (EMSAs) (Supplementary Fig. 1). Under these conditions, nucleotide-free RAD-51-ssDNA complexes mainly form structures observed in the wells of the gel after crosslinking, suggesting such protein-DNA complexes have a tendency to aggregate. In the presence of RFS-1/RIP-1, these are cleared in a concentration-dependent manner, leading to both an accumulation of free ssDNA as well as some intermediate molecular weight protein-ssDNA complexes. Again, this effect was reduced for the mutants relative to the wild-type. Specifically, in the presence of wild-type RFS-1/RIP-1, more of the aggregates were converted to free DNA, whereas with K56A or E138A RFS-1 mutants, less free DNA was observed and more protein-ssDNA structures persisted in the middle of the gels (Supplementary Fig. 1, compare lanes 5, 8 and 11). These results show this activity is intrinsic to the RFS-1/RIP-1 complex and is partially dependent on the RFS-1 Walker boxes.

**Control of RAD-51 ssDNA binding by RFS-1/RIP-1 is species-specific.** To confirm the functional interaction between nematode RAD-51 and RFS-1/RIP-1 is species-specific, we monitored the influence of RFS-1/RIP-1 on human RAD51 (hsRAD51) and yeast Rad51 (scRad51) under different nucleotide co-factor conditions (Supplementary Fig. 2). As expected, RFS-1/RIP-1 failed to stimulate human RAD51 or yeast Rad51 binding to ssDNA in the presence of ATP or ADP (Supplementary Fig. 2a-c, e-g) and also failed to inhibit binding of human RAD51 or yeast Rad51 to ssDNA in the absence of nucleotide co-factor (Supplementary Fig. 2d, h), confirming the RFS-1/RIP-1 control of RAD-51 ssDNA binding requires species-specific interactions.

**Direct RAD-51 interaction with RFS-1 is important for the nucleotide cofactor-dependent control of RAD-51 binding to ssDNA.** We previously defined critical residues mediating the RAD-51/RFS-1/RIP-1 interaction network by mutational yeast two-hybrid analysis. Specifically, we found the D238A mutation in RAD-51 and the L134P mutation in RFS-1 both completely abolished the RAD-51/RFS-1 interaction, in the presence or absence of RIP-1[6]. We considered the possibility that this interaction interface needs to be functional for RFS-1/RIP-1 to block RAD-51 binding to ssDNA in the absence of nucleotide cofactor. We therefore expressed and purified RAD-51 D238A and RFS-1 L134P/RIP-1 for use in the stopped flow (SF) assays described above. A profound defect was observed with both mutants in the ability of RFS-1/RIP-1 to block RAD-51 binding to ssDNA in the absence of nucleotides (Supplementary Fig. 3a-b, e-f). In addition, the use of either mutant impaired the stimulatory effect of RFS-1/RIP-1 on RAD-51 binding to ssDNA in the presence of ATP (Supplementary Fig. 3c, g) or ADP (Supplementary Fig. 3d, h). These data show the direct protein-protein interaction between RAD-51 and RFS-1 is important for these biochemical activities.

To provide some mechanistic insight into how these distinctive functions for RFS-1/RIP-1 in controlling RAD-51 might be regulated by the same protein-protein interaction interface, we performed immunoprecipitation experiments using the C-terminal RIP-1 3xFLAG tag under different nucleotide conditions. We confirmed a direct protein interaction between RAD-51 and RFS-1/RIP-1 protein complex in all nucleotide conditions (Supplementary Fig. 3i). Intriguingly, in the presence of ssDNA and ATP, we observed a weakening of this interaction. In contrast, in the absence of DNA the interaction was reduced slightly in the presence of ADP (Supplementary Fig. 3j). Together, these results suggest the mode of the direct RFS-1/RIP-1 protein-protein interaction with RAD-51 may be regulated by nucleotide, which may partly account for how RFS-1/RIP-1 exhibits different functions under different nucleotide conditions. For example, the

weaker interaction with ATP and ssDNA may reflect the transitory protein-protein interaction observed during filament growth in single-molecule studies[18].

**RFS-1/RIP-1 destabilizes RAD-51-ssDNA complexes containing ADP as a nucleotide cofactor.** We have previously observed that RFS-1/RIP-1 binding to RAD-51-ssDNA complexes strongly reduces the dissociation rate of RAD-51 from ssDNA in the presence of ATP[6]. We were able to detect this effect using a different SF reaction setup, in which pre-formed RAD-51 filaments on Cy3-labelled ssDNA (with or without RFS-1/RIP-1) are challenged with a large excess of unlabeled competitor oligonucleotides. Filament stabilization manifests as a less dramatic Cy3 fluorescence reduction upon incubation with scavenger DNA when RFS-1/RIP-1 is included in the experiment[6].

We therefore used the same reaction setup to test if RFS-1/RIP-1 influences the stability of RAD-51-ssDNA complexes pre-formed in the presence of ADP. RAD-51 was pre-incubated with internally Cy3-labelled ssDNA (oligonucleotides labelled at other positions are described below), ADP and different concentrations of RFS-1/RIP-1, before mixing with 100-fold excess of competitor DNA in the stopped-flow instrument. Remarkably, we observed a strong, concentration-dependent destabilization of the RAD-51-ssDNA complexes by RFS-1/RIP-1 (Fig. 3a, b). The concentration of RFS-1/RIP-1 to reach the half-change of maximal destabilization is 26 nM, corresponding to almost 1/40th of RAD-51 concentration and indicating this activity occurs at highly substoichiometric ratios. We validated this effect using EMSA, which demonstrated the protein-ssDNA complexes formed in the presence of ADP were more prone to dissociation by a range of concentrations of competitor DNA in the presence of RFS-1/RIP-1 (Fig. 3c, d).

**Validation of antagonistic roles of RFS-1/RIP-1 on RAD-51-ssDNA filament stability in the presence of ATP or ADP using bio-layer interferometry.** To further support our observations, we took advantage of an optical analytical method called bio-layer interferometry (BLI), which is a label-free method for measuring biomolecular interactions. BLI monitors the interference pattern of white light reflected by two surfaces, in this case an internal reference layer and a layer of immobilized ssDNA on the biosensor. Any change in the number of molecules bound to the biosensor (e.g., due to protein binding to ssDNA) changes the interference pattern, which can be measured in real-time as a change of 'optical thickness'[19]. In all experiments, 43mer oligo dT was used as the ssDNA substrate, as in SF. However, rather than being modified with a fluorophore, the DNA is biotinylated at the 3′ end, and then conjugated to a streptavidin-coated biosensor.

First, we confirmed that RAD-51 binds the biosensor only in the presence of ssDNA, whereas RFS-1/RIP-1 does not bind directly in either condition (Supplementary Fig. 4a), consistent with prior results from EMSA, stopped-flow and single-molecule FRET[6]. Next, we performed titration experiments where increasing amounts of RAD-51 were bound to DNA in the presence of ATP or ADP. These experiments allowed us to determine the RAD-51 concentration where the ssDNA on the biosensor is completely coated by protein to be 1 μM under these conditions (Supplementary Fig. 4b, c). We also performed an experiment where increasing concentrations of RFS-1/RIP-1 were mixed with this constant saturating concentration of RAD-51 in the presence of ATP. We observed a reproducible decrease in the half-time of the protein-DNA association phase in the presence of RFS-1/RIP-1, suggesting accelerated binding to ssDNA (Supplementary Fig. 4d). These results are consistent with the modest stimulation

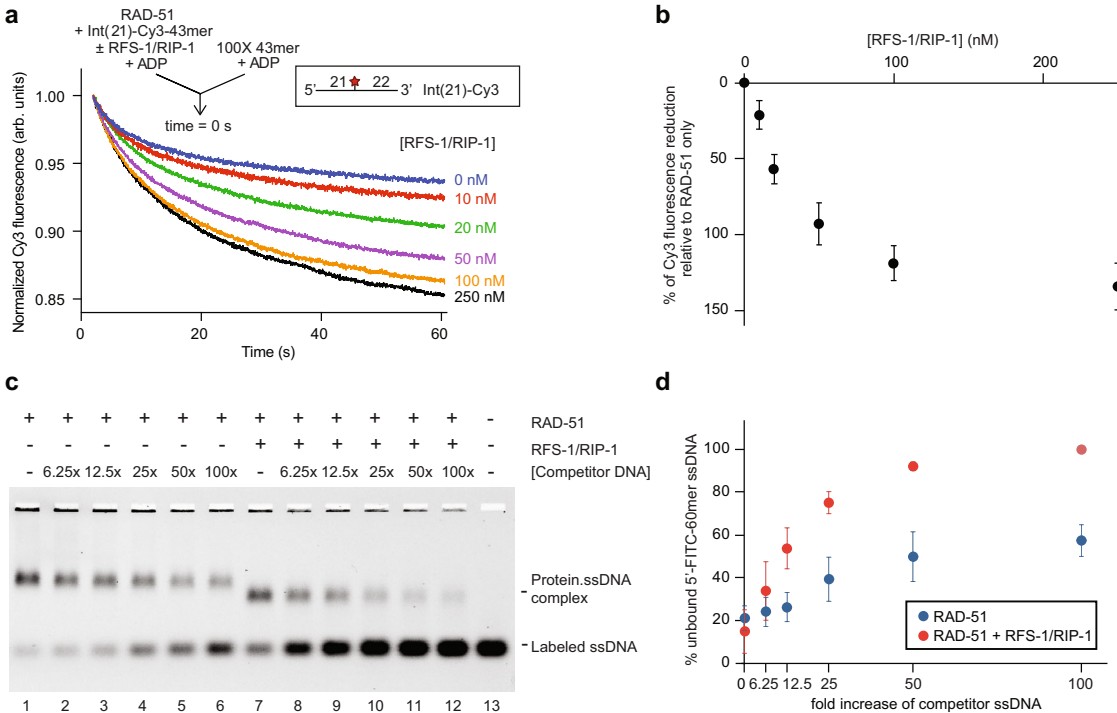

**Fig. 3 RFS-1/RIP-1 destabilizes RAD-51-ssDNA filaments in the presence of ADP. a** Average normalized Cy3-43mer fluorescence profiles plotted as a function of time. The arrow indicates the components of the two syringes rapidly mixed at the 0 s time point in a stopped flow instrument. RAD-51-ssDNA filaments pre-formed with RAD-51 (1 μM) and Int(21)-Cy3 labelled 43mer ssDNA (15 nM) and indicated concentrations of RFS-1/RIP-1 were mixed for 10 min in the presence of ADP. The mixture was then challenged with 100-fold excess of unlabelled 43mer and analyzed (*n* = 5–8). **b** Graph of RFS-1/RIP-1 concentration-dependence change of Cy3 fluorescence from (**a**) (mean; errors: s.d.). **c** Representative EMSA gel (*n* = 4) demonstrating destabilization of RAD-51-ssDNA filaments in the presence of ADP. RAD-51 (1 μM) and RFS-1/RIP-1 (0.5 μM) were pre-incubated before addition of 5′-FITC-61mer ssDNA (10 nM) for 10 min and then challenged with increasing amounts of unlabelled 61mer ssDNA for a further 10 min. Protein-DNA complexes were crosslinked and resolved in agarose gels. **d** The amount of unbound 5′-FITC-61mer ssDNA from (**c**) (*n* = 4) was quantified. The average percentage of unbound 5′-FITC-61mer ssDNA was plotted as a function of the relative excess concentration of unlabelled 61mer over 5′-FITC-61mer ssDNA (mean; errors: s.d.). Source data are provided as a Source Data file.

of saturating RAD-51 binding to ssDNA by RFS-1/RIP-1 observed using SF at saturating RAD-51 concentrations[6], and established conditions where the ssDNA is fully coated by protein for studying RAD-51 dissociation using this new assay.

Next, we utilized this method to confirm the stabilization effect of RFS-1/RIP-1 on RAD-51-ssDNA filaments in the presence of ATP. RAD-51 with or without RFS-1/RIP-1 in the presence of ATP was bound to the ssDNA (association step) and then the filaments were challenged by an excess of free ssDNA (dissociation step) (Supplementary Fig. 5a). In agreement with our previously published stopped-flow data[6], the dissociation of protein molecules bound to the biosensor increased when RFS-1/RIP-1 was omitted, confirming RFS-1/RIP-1 stabilizes RAD-51-ssDNA in the presence of ATP. However, the effect was more modest in this assay under the tested conditions. Since BLI enabled us to monitor the dissociation step in alternative conditions where no competitor DNA is required to promote disassembly of filaments, we also used a variation of the assay in which we increased the concentration of salt to 300 mM (sixfold increase) to initiate the dissociation phase. Under these conditions we could again detect the stabilization effect of RAD-51 filaments by RFS-1/RIP-1 more clearly (Supplementary Fig. 5b), showing this effect is robust to variations in the assay set-up. Finally, we tested the influence of RFS-1/RIP-1 on filament stability in the BLI assay in the presence of ADP. In contrast to ATP, RFS-1/RIP-1 stimulated RAD-51 dissociation under these conditions (Supplementary Fig. 5c), similarly, to stopped-flow experiments (Fig. 3).

In summary, our data from three distinct methods (stopped-flow, EMSA and BLI) reveal that RFS-1/RIP-1 promotes the turnover of the ADP-bound RAD-51-ssDNA filaments, which contrasts with the striking filament stabilization observed in the presence of ATP.

**ADP-RAD-51-ssDNA filament destabilization by RFS-1/RIP-1 is non-polar and partially dependent on the RFS-1 Walker boxes.** The magnitude of ATP-RAD-51-ssDNA filament stabilization by RFS-1/RIP-1 exhibits a striking polarity with respect to the proximity of the position of the Cy3 fluorophore being monitored (a surrogate for different positions within the RAD-51 filament) to RFS-1/RIP-1 bound at the 5′ filament end[7]. We therefore asked whether the filament destabilization activity of RFS-1/RIP-1 toward ADP-RAD-51-ssDNA complexes is similarly polar by performing stopped-flow experiments using 5′, internally, and 3′ labelled oligonucleotides. Notably, the extent of protein-ssDNA complex destabilization in the presence of two different concentrations of RFS-1/RIP-1 was very similar across all three ssDNA constructs (Fig. 4a-d), showing this effect is observed with equivalent magnitude throughout the RAD-51-ssDNA filament and is therefore not polar with respect to the underlying DNA.

Our prior studies also revealed that mutations in the Walker boxes of RFS-1 (K56A and E138A) impede the ability of RFS-1/RIP-1 to stabilize ATP-RAD-51-ssDNA filaments in the SF assay[6]. Under ADP conditions at sub-stoichiometric concentrations of RFS-1/RIP-1 (100 nM), we also observed defects of these

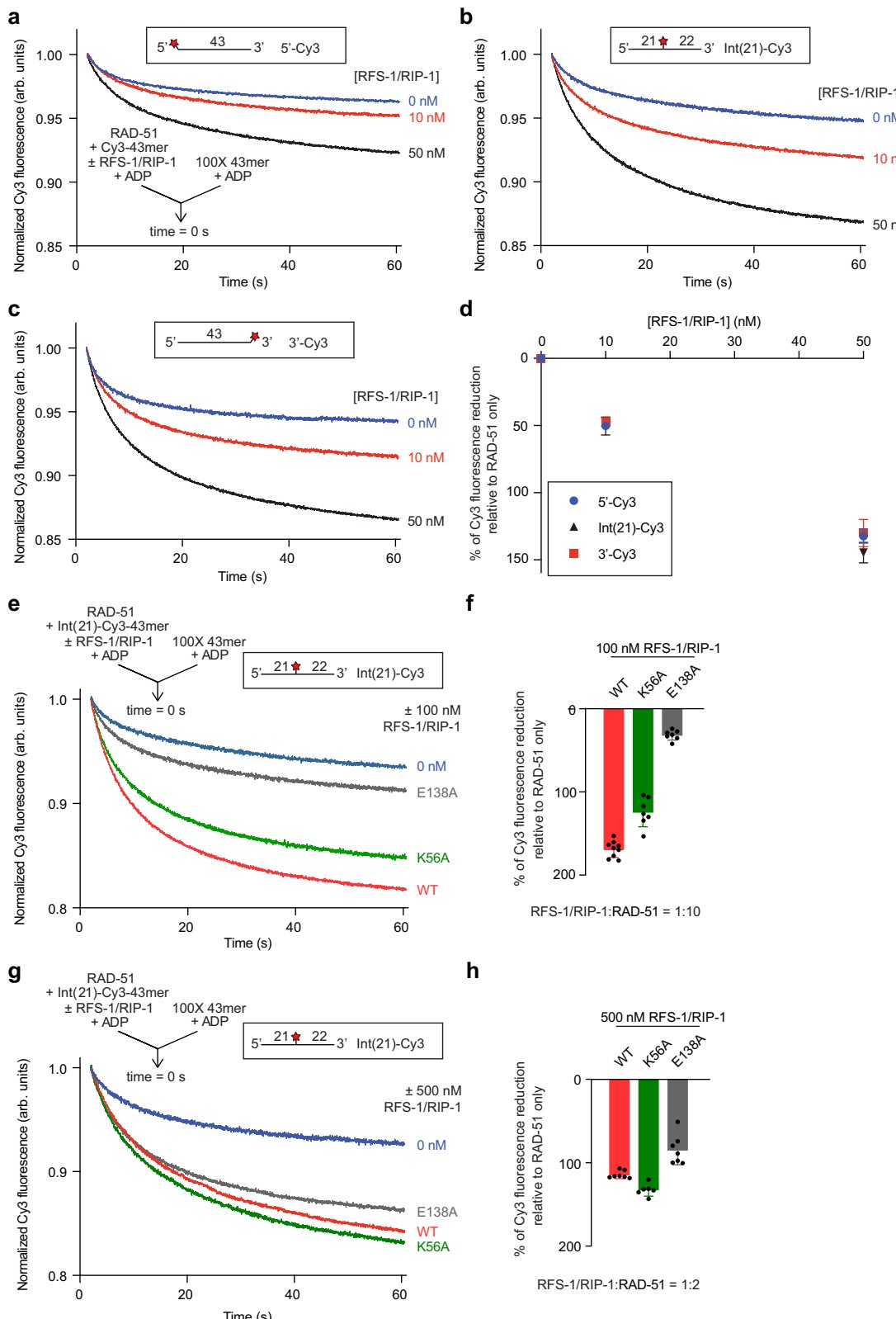

mutants in ADP-RAD-51-ssDNA complex destabilization, particularly for E138A (Fig. 4e, f). However, at higher concentrations (500 nM), the mutants behaved similarly to the wild-type RFS-1/RIP-1 complex (Fig. 4g, h).

While the defects observed for the RFS-1 mutants in ADP-RAD-51-ssDNA destabilization and prevention of nucleotide-free RAD-51 binding to DNA were not comparable to the strong defects these mutants exhibit toward ATP-RAD-51-ssDNA filaments[6], the intermediate defect is qualitatively similar in both cases. The moderate defects in these two biochemical activities argue these functions are intrinsic and specific to the RFS-1/RIP-1 protein complex but may also involve other structural features beyond the Walker boxes of RFS-1/RIP-1.

**Fig. 4 RAD-51 destabilization by RFS-1/RIP-1 occurs throughout the whole filament and is partially dependant on the RFS-1 Walker boxes. a–c** Average normalized Cy3-43mer fluorescence profiles plotted as a function of time. The arrow indicates the components of the two syringes rapidly mixed at the 0 s time point in a stopped flow instrument. Schematics of the different Cy3 label positions are shown inset. RAD-51-ssDNA filaments pre-formed with RAD-51 (1 μM), Cy3-43mer ssDNA (15 nM) and indicated concentrations of RFS-1/RIP-1 in the presence of ADP for 10 min were mixed with 100-fold excess unlabelled 43mer. (**a**) 5′-, (**b**) Int(21) or (**c**) 3′-end Cy3 labelled substrate was used (**a**: n = 17–20, **b**: n = 14–16, **c**: n = 15–16). **d** Graph of RFS-1/RIP-1 concentration-dependence of Δ Cy3 fluorescence for the data presented in (**a–c**) (mean; errors: s.d.). **e–h** Comparison of WT and Walker box mutants of RFS-1 (E138A and K56A) using two different concentrations of RFS-1/RIP-1: 100 nM (**e**) (n = 7–9) and 500 nM (**g**) (n = 4–7). Experiments were done similarly as in **a–c**. **f**, **h** Quantification of the data over the full-time course of the experiments in (**e** and **g**) respectively (mean; errors: s.d.). Source data are provided as a Source Data file.

**RFS-1/RIP-1 acts as a sensor of RAD-51 filament nucleotide content**. The data presented thus far suggest that the RAD51 paralogs stabilize the ATP-bound RAD51-ssDNA pre-synaptic filament but destabilize the non-productive ADP-bound form. It is known that RAD51 undergoes moderate ATP hydrolysis when bound to ssDNA[9–11,20]. Hence, ATP-RAD-51-ssDNA will slowly convert to ADP-RAD-51-ssDNA over time, passing through a population of filaments with mixed nucleotide content. We therefore hypothesized that RFS-1/RIP-1 could act as a sensor of these mixed filaments, such that at a critical ATP:ADP ratio, RFS-1/RIP-1 could exhibit switch-like behavior between stabilization/destabilization activities, driving turnover of filaments at a critical time-out point when ATP content falls too low.

To test if such an ATP:ADP threshold existed, we returned to the BLI method to monitor the dissociation of RAD-51 filaments induced by competitor ssDNA at various ATP:ADP ratios. At the highest ATP:ADP ratios, the protein-DNA complexes were the most stable in the presence of RFS-1/RIP-1 (Fig. 5a, b: blue and red traces). However, at ratios below 3:1, the behavior changed dramatically, and RFS-1/RIP-1 destabilized the filaments (Fig. 5a, b: gray and black traces). Analysis of the overall optical thickness change, representing protein dissociation from the DNA, highlighted this abrupt transition in RFS-1/RIP-1 behavior (Fig. 5c, d), and suggested that the RFS-1/RIP-1 activity switch is triggered at ~0.37 mM ADP (Fig. 5d), representing ~4.4-fold more ATP than ADP in the filament.

Together, our data reveal that RFS-1/RIP-1 serve a nucleotide proofreading function, which stabilizes productive ATP-bound RAD-51-ssDNA complexes, whilst destabilizing non-productive ADP-bound filaments even with relatively limited amounts of ADP present compared to ATP.

## Discussion

The biochemical properties of RAD51 have long been known to be strongly dependent on nucleotide co-factors. This is exemplified in terms of DNA binding and recombinase activities as well as the three-dimensional structures of the nucleoprotein filament[8–10,21]. Broadly, ATP-bound RAD51 on ssDNA is considered to be the biochemical form required for homology search and strand exchange across eukaryotic species including yeast[9], nematode[6] and humans[10]. In contrast, ADP-bound or nucleotide-free RAD51 is deficient for such activities in these organisms[6,9,10]. Similar observations have been made for bacterial[22] and phage[23] recombinases.

In addition, HR and RAD51 activities are tightly regulated in eukaryotic organisms. Positive regulators ensure efficient progression through the complex reaction and limit the accumulation of broken DNA, joint molecule intermediates and stalled replication forks, which pose severe risks to genome stability if allowed to persist. Conversely, negative regulators reverse the HR reaction when it is initiated inappropriately[24]. Many of these regulatory activities act at the level of the RAD51-ssDNA filament, in terms of its formation and stability. We and others have previously shown that nematode and human BRCA2 proteins

stimulate RAD51 binding to DNA (in the presence or absence of RPA) and also inhibit ATP hydrolysis by RAD51[2,20,25]. In this way BRCA2 is a strong positive regulator of early HR, driving formation and prolonging the lifetime of the ATP-bound, active pre-synaptic complex.

RAD51 paralogs also regulate RAD51, but have been challenging to study for many years due to their tendency to form protein-protein and protein-DNA aggregates in biochemical studies[26]. However, co-expression of RAD51 paralogs facilitated study of their activities. Surprisingly, RAD51C-XRCC3 and RAD51D-XRCC2 human paralog complexes were shown to promote D-loop formation in vitro[27,28]. Similarly, the RAD51B-RAD51C subcomplex has also been shown to have in vitro RAD51 mediator activity[29]. Our studies of the RFS-1/RIP-1 complex from nematodes also enabled us to overcome many technical challenges in the field: this allowed us to elucidate its function in stabilizing and remodeling RAD-51 filaments on ssDNA, prolonging the lifetime of the pre-synaptic complex and stimulating recombinase activity[6].

In the present study, we show that RFS-1/RIP-1 stimulates binding to ssDNA in the presence of nucleotide co-factors. However, in this context, RFS-1/RIP-1 cannot discriminate between the inactive ADP-bound form of RAD-51 and the active ATP-bound version. This raised the possibility that ADP-bound RAD-51 may accumulate inappropriately on ssDNA and that a regulatory activity limiting this might exist. Indeed, we show that RFS-1/RIP-1 confers a profound destabilizing effect on ADP-RAD-51-ssDNA complexes, in striking contrast to its effect on the ATP-bound form. In addition, this may serve to ensure turnover of RAD-51-ssDNA complexes which have started to undergo ATP hydrolysis and convert toward the inactive ADP-bound form. We also show that RFS-1/RIP-1 demonstrates switch-like behavior between its stabilizing and destabilizing functions when ADP content of the filament exceeds ~25%. Finally, we also demonstrated that RFS-1/RIP-1 prevents any binding of RAD-51 in a nucleotide-free state to DNA and established a new BLI-based method to study behavior of RAD51 nucleoprotein filaments. Together, this array of 'nucleotide proofreading' activities within the RFS-1/RIP-1 complex, controlling RAD-51 DNA binding and stability with exquisite nucleotide sensitivity, promote the specific accumulation and persistence of the active, ATP-bound form of RAD-51 on ssDNA (Fig. 5e).

Intriguingly, our observations raise the possibility that ADP-RAD-51-ssDNA complex destabilization by RFS-1/RIP-1 may at least in part occur via a distinct mechanism of action to its ability to stabilize ATP-RAD-51-ssDNA filaments, since the former activity is observed throughout the filament (without any polarity) and is less critically dependent on the RFS-1 Walker boxes in the SF assay. RFS-1/RIP-1 may exert some of its proofreading activity via sensing of nucleotide status at the filament ends. For example, ADP-RAD-51-ssDNA complexes may be more discontinuous than ATP-bound forms, making more ends available for RFS-1/RIP-1 to access, explaining the lack of polarity but also

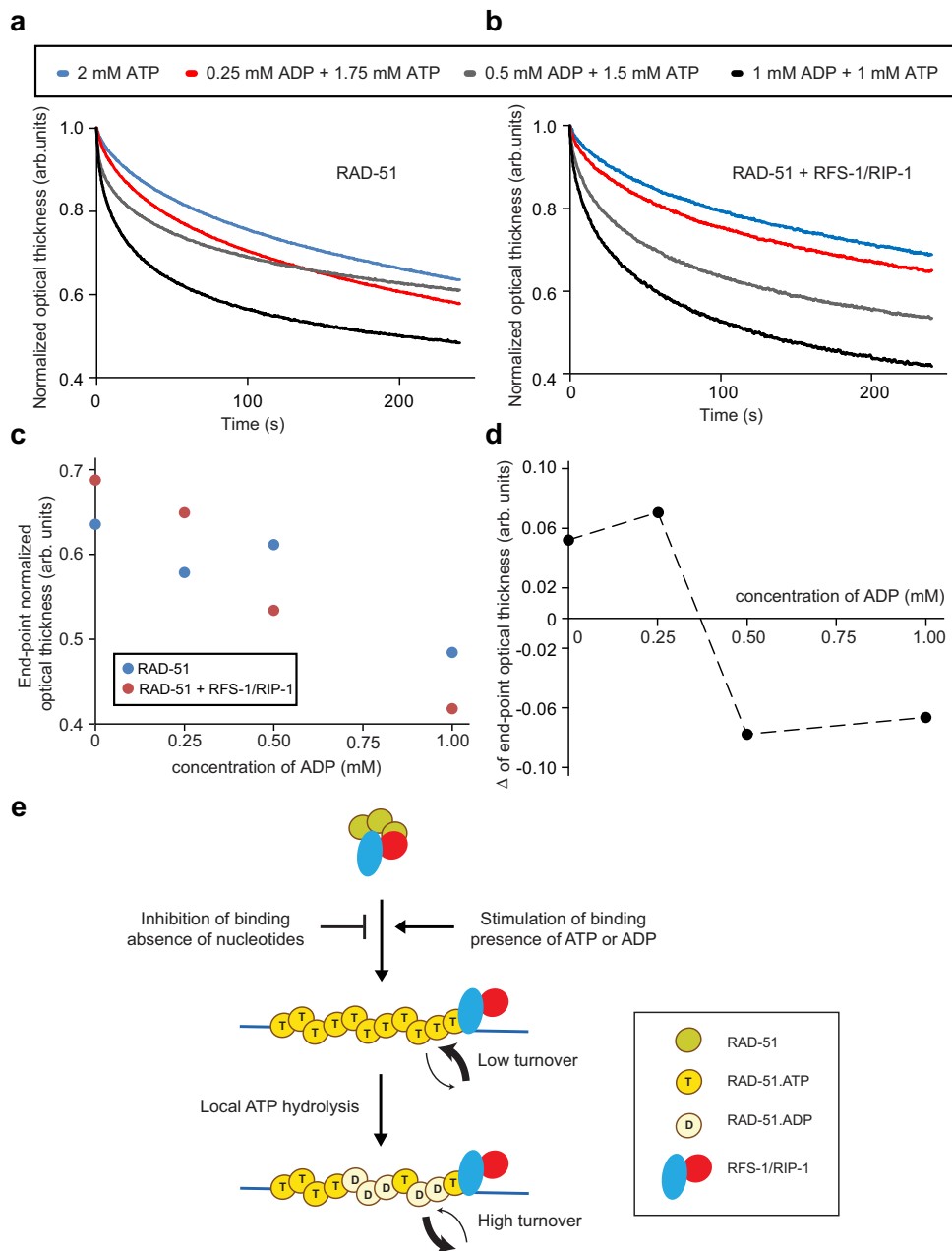

**Fig. 5 A threshold of ADP in the RAD-51-ssDNA filament triggers a switch in RFS-1/RIP-1 activity from stabilization to destabilization. a, b** Normalized dissociation phases obtained from BLI experiments using 3′ biotinylated ssDNA (15 nM) where RAD-51 (1 μM) alone (**a**) or in the presence of RFS-1/RIP-1 (0.25 μM) (**b**) was loaded on the biosensor together with different ATP/ADP ratios. The biosensor with bound proteins was then submerged in buffer without any nucleotide cofactor and 1000-fold excess of unlabelled ssDNA to facilitate dissociation. **c** The corresponding endpoint in optical thickness across the dissociation phase for the data presented in (**a** and **b**) is plotted as a function of ADP concentration. **d** The difference in the values in (**c**) for RAD-51 ±RFS-1/RIP-1 was plotted to estimate the break point where stabilization switches to destabilization. **e** Schematic model of the action of RAD51 paralogs based on our results. RAD-51 binding to ssDNA is inhibited by RAD51 paralogs in the absence of nucleotides, whereas in ATP or ADP the formation of nucleofilaments is stimulated. Filaments undergoing local ATP hydrolysis are destabilized leading to RFS-1/RIP-1-mediated filament turnover. Source data are provided as a Source Data file.

the partial dependence on the RFS-1 Walker boxes, known to help mediate protein-protein interactions with RAD-51[6]. Nevertheless, we cannot exclude that RFS-1/RIP-1 may also partly proofread the RAD-51 filament via alternate modes, for example lateral associations at regions of altered RAD-51 structure at protomers which have converted from an ATP to ADP-bound state.

RAD51 hydrolyzes ATP when bound to DNA, which converts ATP bound between adjacent RAD51 protomers to ADP,

destabilizing the filament[8,25]. Nevertheless, RAD51 can form filaments in both states with similar kinetics and affinities[30]. The overall tendency of ADP-bound or nucleotide-free RAD51 to bind ssDNA in competition with ATP-bound RAD51 in vivo is unclear. However, the ADP-RAD-51-ssDNA destabilizing activity we define is likely to be important in rapidly switching off unproductive filaments once a critical ADP threshold is reached to compromise recombinase or replication fork protection activities (Fig. 5e). Due to relatively rapid ATP hydrolysis and

slow ADP dissociation[31], the ATP-RAD-51-ssDNA filament might be converted into mixed ATP/ADP-RAD-51-ssDNA that is unable to perform strand exchange but stable enough[30,32] to represent such an unproductive filament. ADP content probably needs to exceed 30% to define this state, since this level is not inhibitory for DNA strand exchange for human RAD51[33]. Furthermore, previous single-molecule work has suggested that significant amounts of ATP hydrolysis can occur within RAD51-ssDNA or dsDNA filaments prior to burst-like dissociation events of multiple protomers[32,34]. The ADP-RAD-51-ssDNA destabilizing activity we define may facilitate such mechanisms and the RFS-1/RIP-1 complex might help change the rate-limiting reaction step. Other recombination regulators have been proposed to influence this process distinctively from 'nucleotide proofreading', by directly regulating RAD51 nucleotide content. For example, SWI5-SFR1 has been shown to enhance ATP hydrolysis and aid ADP release from RAD51-ssDNA[35] and XRCC2 has been proposed to promote ADP-to-ATP exchange within RAD51[36], somewhat analogous to a guanine nucleotide exchange factor in promoting GDP-to-GTP exchange. To switch off the recombinase activity or remove untimely or toxic RAD51 filaments, several negative regulators have been described[37–39]. Among them, the yeast Srs2 anti-recombinase actually triggers ATP hydrolysis within the RAD51 filament to promote its dissociation[40]. It is tempting to speculate whether the paralogs could cooperate with anti-recombinases to make the clearance more efficient in the in vivo setting following ATP turnover.

While RFS-1/RIP-1 clearly has distinctive roles in controlling RAD-51-ssDNA structures in the presence of different nucleotides on naked DNA, it is unclear how this will be influenced by the presence of RPA binding. In the context of long exposed tracts of ssDNA, individual RPA molecules will bind with a footprint of ~30 nucleotides of ssDNA with very high affinity[41], where mediator proteins such as BRCA2 are important for promoting RAD51 binding by driving RPA displacement[42,43]. The mechanisms we reveal may gain greater importance in the context of a stalled replication fork, where RFS-1/RIP-1 is thought to have a major role in vivo[44]. When a replication fork encounters DNA damage, the replicative helicase slows down following stalling of the leading strand polymerase[45], which could lead to a window of opportunity in which a time lag in ssDNA exposure permits RAD51 to bind prior to RPA, since the binding footprint of individual RAD51 protomers is just three nucleotides, tenfold less than the optimal footprint of RPA[46]. In this context, it is also notable that the minimal binding footprint of BRCA2 to ssDNA is at least 40 nucleotides and optimal footprint 74 nucleotides[47]. The RAD51 paralogs thus could have an important role in establishing and maintaining RAD51 in a nucleotide-bound state on short naked ssDNA exposed briefly after fork stalling via the mechanisms we describe here, although the DNA resection machinery is also likely to increase ssDNA exposure at stalled forks, making this potential window transitory. RAD51 has important functions at forks including protection from degradation, promotion of fork reversal and restart[12,15,48–50]. The RAD51 paralogs thus may play a role therein to help enrich the stable ATP-bound form of RAD51 as well as promote succession of individual steps during fork processing/DSB repair. Indeed, the human RAD51 paralog complex BCDX2 has recently been reported to promote replication fork reversal independent of BRCA2, which may reflect a role in facilitating RAD51 binding to ssDNA. This is potentially a context in which ssDNA is exposed and RPA binding is more limited, and may provide a biologically-important window in which the nucleotide proofreading functions of RAD51 paralogs control RAD51 binding to ssDNA to ultimately help regulate replication fork dynamics[51].

Following strand exchange, RAD51 bound to dsDNA must also be removed to allow initiation of DNA repair synthesis[52], which is in part thought to be facilitated by RAD51 ATPase activity. An interesting topic for future study will therefore be to decipher how the activities reported here for RFS-1/RIP-1 extend to RAD-51 bound to dsDNA as well as fork reversal and fork restart. We previously showed that a peptide of RFS-1 that binds RAD-51 can destabilize RAD-51-dsDNA filaments, which acts redundantly with HELQ-1 to facilitate later steps in HR[53]. How this translates to the full-length RFS-1/RIP-1 complex and how this might be influenced by different nucleotide-bound forms of RAD-51 in the context of various DNA substrates will clarify the mechanism underlying this function.

In summary, we have defined 'nucleotide proofreading' mechanisms by which RAD51 paralogs interrogate and respond to the nucleotide content of RAD51. In concert with the previously demonstrated functions in stabilizing RAD51 on ssDNA when bound to ATP and remodeling of RAD51 filaments to an active form, this suite of biochemical effects helps ensure the persistence of active forms of the RAD51 recombinase, providing a further level at which RAD51 paralogs help drive forward the HR reaction. Ultimately, loss of this complement of functions likely underpins the profound genome stability defects and accumulation of mutations associated with defects in RAD51 and its paralogs, including in cancer cells[6,54,55].

## Methods

**Protein expression and purification**. RAD-51 and RFS-1/RIP-1 and corresponding mutants were purified as described previously[6]. Briefly RAD-51 was expressed using the Champion pET-SUMO system in *E. coli* (Life Technologies). Protein was bound to Ni-NTA agarose affinity gel, eluted by imidazole and dialyzed to remove imidazole. His-tagged Ulp1 SUMO protease was used to remove the His-SUMO tag, and Ni-NTA agarose affinity gel added to remove protease. RAD51-51 was further purified by Mono Q column, eluted fractions were pooled, concentrated, and stored at −80 °C. RFS-1/RIP-1 was expressed in budding yeast cells. Cells were crushed in a freezer mill, proteins resuspended in buffer, cleared lysate by using ultra centrifuge was applied to anti-FLAG affinity gel. Beads were extensively washed, and proteins were eluted by addition of 3xFLAG peptide. After dialysis RFS-1/RIP-1 was concentrated and stored at −80 °C. Both RAD-51 and RFS-1/RIP-1 were devoid of detectable nuclease activity.

**Stopped-flow assays and data analysis**. Stopped-flow experiments were done as described in[6]. Briefly experiments were performed using an SFM-3000 stopped-flow machine fitted with a MOS-500 monochromator spectrometer (Bio-Logic) with excitation wavelength set at 545 nm. Fluorescence measurements were collected with a 550 nm long pass emission filter. The machine temperature was maintained at 25 °C with a circulating water bath.

For all experimental setups, a master mix containing all common reaction components for each of the two syringes was prepared to which variable components were added to generate the mixtures for individual syringes for different experimental conditions. These individual syringe mixtures are indicated in the mixing schemes. All concentrations quoted represent final concentrations after mixing. Components of each syringe were pre-incubated for 10 min before the start of experiments to allow the contents to reach equilibrium.

All reactions were performed in SF Buffer (50 mM Tris-HCl (pH 7.5), 5 mM MgCl$_2$, 50 mM NaCl, 2 mM ATP or ADP or absence of nucleotide). All reactions contained 15 nM (moles) Cy3 fluorescently labelled (dT)$_{43}$ oligonucleotide (Cy3-43mer), with the fluorophore either conjugated to the 5′ or 3′ end or integrated into the DNA backbone after the indicated nucleotide position (Supplementary Table 1). Unlabeled (dT)$_{43}$ oligonucleotide was used at 1500 nM (moles) (100-fold excess) in competition experiments. For all experiments, controls were also performed for buffer alone with and without DNA to confirm fluorescence signal stability over the time course of the experiments (data not shown). For experiments on pre-formed RAD-51 filaments we employed a saturating concentration of RAD-51 (1 μM on 43mer), to ensure all ssDNA present was coated by RAD-51 filaments. Fluorescence measurements for most experiments were collected according to the following protocol: (1) every 0.00005 s from 0 to 0.05 s; (2) every 0.0005 s from 0.05 to 0.56 s; (3) every 0.02 s from 0.56 to 60.54 s using BioLogic Bio-kine32 software.

For each condition analyzed, traces were collected from several independent reactions. For presentation, average traces for each experiment were generated. Absolute fluorescence values were converted to arbitrary units by a normalization procedure to facilitate comparison[6]. For analysis, for all experiments ten-point moving averages were calculated on each individual normalized trace, which were used to define initial (0 s) and final (60.54 s) fluorescence values and Δ Cy3

fluorescence values for each experiment. Half-times were measured as the time points where the fluorescence from moving averages was closest to the value calculated for the Δ Cy3 fluorescence midpoint. Average values and associated standard deviations were then calculated. Positive and negative Δ Cy3 fluorescence values represent increases and decreases in fluorescence, respectively. To determine the % of Cy3 fluorescence reduction observed for RAD-51, the Δ Cy3 fluorescence value for each trace calculated in the presence of RFS-1/RIP-1 was divided by the average Δ Cy3 fluorescence value for RAD-51 alone and multiplied by 100. Average values and associated standard deviations or standard errors of the mean were then calculated using Microsoft Excel 365 software.

**Biolayer-interferometry assay**. BLI experiments were performed using a single–channel BLItz instrument in Advanced Kinetics mode or multichannel Octet RED96e (ForteBio) at room temperature and with shaking at 2.200 rpm. Prior to the measurements the streptavidin biosensors (SAX, ForteBio, Cat No. 18-0037) were pre–hydrated by incubation with BLI buffer (SF buffer supplemented with 0.05% Tween 20) for 10 min and loaded with biotinylated dT43 ssDNA (Supplementary Table 1) for 120 s. Data for dissociation phase were normalized to the starting point. The change in signal was then plotted as a function of time. The binding affinity of the proteins to ssDNA was measured as an increase in the thickness of the biomolecule layer in nanometers (nm). For data acquisition we used either ForteBio BLItz Pro or ForteBio Data Acquisition software, Microsoft Excel 365 was used to analyze the data.

**EMSA**. 10 nM FITC-labelled 61mer (Supplementary Table 1) was mixed with either 1 µM RAD-51 or 1 µM RAD-51 plus 0.5 µM RFS-1/RIP-1 in SF buffer at 25 °C. After 10 min, increasing amounts of unlabeled competitor ssDNA of the same sequence was added. Following additional 10 min, the reaction was stopped by crosslinking with glutaraldehyde with final concentration 0.125% and samples were resolved on agarose gel. Gels were imaged on a FLA-9000 scanner (Fujifilm) using Carestream Molecular Imaging Software SE and analyzed by Fujifilm MultiGauge software.

**Reporting summary**. Further information on research design is available in the Nature Research Reporting Summary linked to this article.

## Data availability

The data supporting the findings of this study are available within the article, its Supplementary Information files; and are available from the corresponding author upon reasonable request. Source data are provided with this paper.

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

## Acknowledgements

The L.K. laboratory is supported by Czech Science Foundation (GACR 21-22593X); Masaryk University [MUNI/G/1594/2019]; European Union's Horizon 2020 research and innovation programme under grant agreement No 812829; the National Program of Sustainability II (MEYS CR, project no. LQ1605); and the European Structural and Investment Funds, Operational Programme Research, Development and Education —"Preclinical Progression of New Organic Compounds with Targeted Biological Activity" (Preclinprogress)—CZ.02.1.01/0.0/0.0/16_025/0007381. The S.B. laboratory is funded by the Francis Crick Institute and grants from the Wellcome Trust and European Research Council (RecMitMei). Both L.K. and S.B. are supported by Wellcome Trust Collaborative Grant 206292/E/17/Z. M.R.G.T. was supported by a Sir Henry Wellcome Postdoctoral Fellowship from the Wellcome Trust (110014/Z/15/Z). We thank Joe Yeeles for critical reading of the paper and Core Facility Biomolecular Interactions and Crystallization of CEITEC MU.

## Author contributions

M.S., M.R.G.T., S.J.B. and L.K. conceived the study. M.R.G.T. and M.S. purified proteins. M.R.G.T., M.S. and O.B. performed stopped-flow experiments, M.S. performed EMSA and BLI assays. M.S., M.R.G.T. and L.K. wrote the paper with input and comments from all authors.

## Competing interests

The authors declare no competing interests.
