## [Peer Review File · Nature Communications]

Nucleotide proofreading functions by nematode RAD51 paralogs facilitate optimal RAD51 filament functionREVIEWER COMMENTS

Reviewer #1 (Remarks to the Author):

Assembly and disassembly of RAD51-ssDNA presynaptic filaments are tightly regulated processes in order to maintain genome integrity. The association and dissociation of RAD51 filaments are influenced by their intrinsic ATPase activity and by their interacting partners, such as RAD51 paralogs. Krejci's lab previously demonstrated that the nematode RAD-51 paralog, RFS-1/RIP-1, possesses a critical function in stabilizing RAD-51-ssDNA filaments by attenuating RAD-51 dissociation from ssDNA. However, how the RFS-1/RIP-1 complex modulates the formation of RAD-51 filaments with ATP, ADP, and in the absence of nucleotide had remained unexplored. Here, the authors took various approaches—including stopped-flow, BLI, and EMSA—to address the kinetics of RAD-51 filament assembly. They found that the RFS-1/RIP-1 complex enhances assembly of RAD-51-ssDNA presynaptic filaments in the presence of either ATP or ADP nucleotide cofactors, but that RAD-51 binding to ssDNA is prevented by RFS-1/RIP-1 in the absence of nucleotides. Importantly, they further demonstrate that the RFS-1/RIP-1 complex stabilizes ATP-bound RAD-51-ssDNA presynaptic filaments but destabilizes inactive ADP-bound filaments. Consequently, they propose that 'nucleotide proofreading' activities within the RFS-1/RIP-1 complex extend the lifetime of active RAD-51-ssDNA-ATP presynaptic filaments. In summary, the authors present a systematic and interesting perspective of how a RAD51 paralog influences the dynamics of RAD51 filament formation under different nucleotide conditions. Addressing the following concerns would strengthen the evidence presented in the current manuscript:

1. Is the physical interaction between RAD-51 and the RFS-1/RIP-1 complex in solution and in the presence of ssDNA affected by different nucleotide cofactors (no nucleotide, ADP, or ATP)?
2. Since RFS-1/RIP-1 harbors this nucleotide proofreading activity, does RFS-1/RIP-1 stimulate RAD-51 ATP hydrolysis both in the presence and absence of ssDNA?
3. It will be important to show if RFS-1/RIP-1 modulates RAD-51 association and dissociation on dsDNA by using the same approaches as already described herein, since RAD-51 bound to dsDNA needs to be removed to allow initiation of DNA repair synthesis.
4. Figures 1 and 2. It will be important to include prokaryotic RecA or human RAD51 to establish if the functional interaction between nematode RAD-51 and RFS-1/RIP-1 is species-specific.
5. Figure 3a. Other constructs (3'-Cy3 and 5'-Cy3 ssDNA) should be included to establish if RFS-1/RIP-1 destabilizes RAD-51-ssDNA filaments in the presence of ADP.
6. Supplementary Fig. 2c & d. It is important to include error bars for these experiments to conclude that co-incubation of saturating RAD-51 with RFS-1/RIP-1 does reveal a slight increase in RAD-51 binding to DNA in the presence of both ATP and ADP.
7. Supplementary Fig. 3. Quantitative analysis and error bars for these experiments are needed.
8. Page 14. "We previously showed RAD-51-ssDNA filaments...." should be "RAD-51-dsDNA filaments".

Reviewer #2 (Remarks to the Author):

Spirek et al report novel attributes of the nematode RAD51 paralog complex RFS1/RIP1 as a molecular switch capable of monitoring and responding to the nucleotide-associated states of RAD51-ssDNA nucleoprotein filaments. Using stop-flow methods and bio-layer interferometry, they have analyzed how RFS1/RIP1 affects RAD51-ssDNA filament dynamics in the presence of ATP and ADP. The main findings are 1) RFS1/RIP1 stimulates RAD51-ssDNA filament formation, 2) RFS1/RIP1 destabilizes ADP-bound RAD51-ssDNA, and thus favors the ATP-bound catalytically active RAD51 filament state, and 3) RFS1 Walker mutants are impaired for this functional

attribute. Moreover, the authors provide evidence that the activity of RFS1/RIP1 is responsive to the ATP/ADP ratio.

Based on the results, the authors propose that RFS1/RIP1 fulfills a 'nucleotide proofreading' role in RAD51 filament maintenance. The work has been done well and significantly broadens our understanding of the mechanism of the conserved RAD51 paralogs on RAD51 filament assembly, activity, and maintenance.

There are a few concerns that need to be addressed.

Specific comments

1. Figure 1 shows relative fluorescence changes for 5', 3' or internally Cy3-labelled ssDNA as being different in ATP vs ADP experiments. Do these differences simply reflect experimental variations, or do they indicate differential effects of RAD51 paralog action with respect to fluorophore position?
2. In Fig 2A, how RFS1/RIP1 inhibits RAD51 binding to ssDNA in the absence of nucleotide remains unclear. Is it governed by direct interaction between RAD51 and RFS1? It will be informative to examine if mutants of RAD51 impaired for RFS1 interaction (Taylor et al, Cell, 2015) but proficient in DNA binding in the same analysis.
3. The data in Fig 2C are inconclusive. DNA binding assays like in Fig S1 or an alternative approach should be employed to support these data.
4. In Fig 2E, like WT, the K56A and E138A RFS1 mutants can also inhibit RAD51 binding to ssDNA in the absence of nucleotide. Since these mutants retain interaction with RAD51 (Taylor et al 2015), It will be interesting to examine if RFS1 mutants lacking RAD51 interaction as described in the previous study fail to inhibit Rad51 binding to ssDNA in the absence of nucleotide.
5. In Fig 4, the magnitude of destabilization of RAD51-ssDNA filaments (% fluorescence reduction, 4d) by RFS1/RIP1 in the presence of ADP was comparable, but the kinetic changes in 5' - vs 3'- or internal labeled probe were different (Fig 4A-C). Please explain.
6. On page 14, while discussing the potential role of RFS1/RIP1 at stalled replication forks by suggesting priority binding of Rad51 over RPA at very narrow stretches of ssDNA, the model should be discussed within the context of known role of the DNA resection machineries.

Reviewer #3 (Remarks to the Author):

In this manuscript, Špírek et al have analyzed how RFS-1/RIP-1 (a nematode RAD51 paralog complex) influences the interaction of RAD-51 with single-stranded DNA (ssDNA) using in vitro "kinetic" assays including rapid-mixing:stopped-flow and Bio-Layer Interferometry. They find that RFS-1/RIP-1 enhances RAD-51 association with ssDNA when experiments are done in the presence of ATP or ADP. However, in absence of nucleotides RFS-1/RIP-1 has the opposite effect, inhibiting RAD-51 association with ssDNA. In terms of effects on RAD-51 dissociation, the authors find that RFS-1/RIP-1 stabilizes RAD-51/ssDNA complexes when ATP is present but promotes dissociation in the presence of ADP. They report that these RFS-1/RIP-1 effects (slow association without nucleotides and faster dissociation of ADP coordinated RAD-51 complexes) occur at sub-stoichiometric ratios relative to RAD-51 monomer concentrations, are not polar, and partially dependent on Walker motifs in RFS-1/RIP-1. The authors propose that RFS-1/RIP-1 has nucleotide proofreading functions.

The idea of "nucleotide proofreading" promoted by the worm RAD51 paralog complex is interesting and will open experimental routes to better understand the molecular functions of RAD51 paralog complexes in vertebrates. The manuscript is well documented and pleasant to read.

Comments and suggestions:

- 1) The title should say that the findings of this work concern the nematode RAD51 paralog complex.
- 2) The first paragraph of the results (p 3) is difficult to understand. Could it be rephrased?
- 3) Figure 1. What is meant by n= 4-5

- 4) Figure 2a and b should be combined with Figure 1 and should be done at the same RAD-51 concentration
- 5) Figure 2 legend: "(c)." no period – "as in A" as in a
- 6) Sup Fig 2c. RFS-1/RIP-1 does not appear to have major effects on association rates. Could you comment.
- 7) Sup Figure 2d; color code is not clear
- 8) BLI experiments of Sup Figure 3 are nice and should be a main figure. Rates should be given.
- 9) About the quantifications: Could information be retrieved on the ON and OFF rates rather than by reporting the differences between fluorescence end points?
- 10) Discussion: Strand exchange by RAD51C-XRCC3 and RAD51B-RAD51C. Was this assessed by D-loop assays? If yes it should be noted that D-loop products do not necessarily form through "a strand exchange mechanism" but could be the results of strand annealing activity dependent on the quality of the dsDNA used in the assay.
- 11) A little weakness of the work is the lack of tests in cells for the "proofreading" activity of RFS-1/RIP-1.

RESPONSE TO REVIEWER COMMENTS

We would like to thank all reviewers for their valuable comments and suggestions that helped to improve our manuscript.

Reviewer #1 (Remarks to the Author):

Assembly and disassembly of RAD51-ssDNA presynaptic filaments are tightly regulated processes in order to maintain genome integrity. The association and dissociation of RAD51 filaments are influenced by their intrinsic ATPase activity and by their interacting partners, such as RAD51 paralogs. Krejci's lab previously demonstrated that the nematode RAD-51 paralog, RFS-1/RIP-1, possesses a critical function in stabilizing RAD-51-ssDNA filaments by attenuating RAD-51 dissociation from ssDNA. However, how the RFS-1/RIP-1 complex modulates the formation of RAD-51 filaments with ATP, ADP, and in the absence of nucleotide had remained unexplored. Here, the authors took various approaches—including stopped-flow, BLI, and EMSA—to address the kinetics of RAD-51 filament assembly. They found that the RFS-1/RIP-1 complex enhances assembly of RAD-51-ssDNA presynaptic filaments in the presence of either ATP or ADP nucleotide cofactors, but that RAD-51 binding to ssDNA is prevented by RFS-1/RIP-1 in the absence of nucleotides. Importantly, they further demonstrate that the RFS-1/RIP-1 complex stabilizes ATP-bound RAD-51-ssDNA presynaptic filaments but destabilizes inactive ADP-bound filaments. Consequently, they propose that 'nucleotide proofreading' activities within the RFS-1/RIP-1 complex extend the lifetime of active RAD-51-ssDNA-ATP presynaptic filaments. In summary, the authors present a systematic and interesting perspective of how a RAD51 paralog influences the dynamics of RAD51 filament formation under different nucleotide conditions. Addressing the following concerns would strengthen the evidence presented in the current manuscript:

1. Is the physical interaction between RAD-51 and the RFS-1/RIP-1 complex in solution and in the presence of ssDNA affected by different nucleotide cofactors (no nucleotide, ADP, or ATP)?

This is an interesting question which probes at the underlying nucleotide proofreading mechanism and whether this is linked to altered protein-protein interactions in different nucleotide co-factor states. This has been somewhat challenging to address historically due to the limited quality of antibodies to enable immunoprecipitation of RAD-51, RFS-1 and RIP-1 directly in the presence of different nucleotides or ssDNA. As such our prior studies to demonstrate the interaction network between these proteins have mainly focused on demonstrating the binding of the RFS-1/RIP-1 complex to RAD-51-ssDNA filaments by EMSA, electron microscopy and stopped flow analysis in the presence of ATP, as well as yeast two-hybrid experiments.

To address the reviewer's specific question, we have utilized the 3xFLAG tag at the C-terminus of RIP-1 to perform immunoprecipitation experiments (Supplementary Figure S3i). We mixed RAD-51 and RFS-1/RIP-1 complex in the presence of ATP or ADP with anti-FLAG M2 agarose beads, washed, and eluted bound proteins by boiling in Laemmli buffer. We observed that RFS-1/RIP-1 binds to RAD-51 more efficiently in the absence of nucleotides (lane 6) compared to in the presence of ATP (lane 7) or ADP (lane 8). A similar result was observed in the presence of ssDNA (lanes 12-14). The difference in levels of RAD-51 pulled down under these various conditions suggests the physical interaction with RFS-1/RIP-1 is influenced by nucleotide cofactor. We speculate in the revised manuscript (page 7) that this result may reflect a role for RFS-1/RIP-1 in physically sequestering RAD-51 from ssDNA in the absences of nucleotides, while weaker or more transient interactions may promote filament growth in the presence of ATP or ADP, as also suggested by the single molecule studies of Belan *et al*, Molecular Cell, 2021.

Together, these results show a direct protein interaction between RFS-1/RIP-1 and RAD-51, support the view that the mode of interaction is distinct under different nucleotide conditions, and could provide part of the explanation for the differential modes of action of RFS-1/RIP-1 observed. While this is an intriguing insight, pull-down experiments only offer a glimpse of the protein-protein interaction network at equilibrium. However, more quantitative and mechanistic information is likely to require sophisticated single-molecule real-time analysis, which is beyond the scope of this study. We have built a summary of the above data and conclusions into the revised manuscript (page 7 and Supplementary Figure S3i).

2. Since RFS-1/RIP-1 harbors this nucleotide proofreading activity, does RFS-1/RIP-1 stimulate RAD-51 ATP hydrolysis both in the presence and absence of ssDNA?

This is an intriguing question. We have attempted to assess the relationship between the ATPase activities of RAD-51 and RFS-1/RIP-1, however we have not observed any effect of RFS-1/RIP-1 on RAD-51 ATPase activity (Figure R1). Nevertheless, the experiments are challenging to interpret for a number of reasons. RAD-51 is a relatively weak ATPase compared to other recombinase enzymes such as *E. coli* RecA, and we have been unable to detect clear ATPase activity of RFS-1/RIP-1 alone either in the presence or absence of ssDNA (Taylor *et al*, Cell, 2015: Figure S2). This combination creates a very limited dynamic range to interpret results from mixing the proteins. More importantly, it is also impossible to assign ATPase activities to RAD-51 or the RFS-1/RIP-1 complex (or both) in experiments where the proteins are mixed, given both harbor ATPase motifs. This in turn is difficult to dissect with Walker A or B box mutants, since mutations in these residues also impact the protein-protein interaction network as we showed previously using yeast two-hybrid experiments (Taylor *et al*, Cell, 2015: Figure S6E-F), and are therefore not clean separation of function mutants. For these reasons, coupled with the fact we have not made conclusions on the interaction between any ATPase functions of either RAD-51 or RFS-1/RIP-1 and proofreading functions in the present study, we do not believe that the inclusion of ATPase experiments are relevant or interpretable without major technical caveats.

Figure R1. Effect of RFS-1/RIP-1 on ATP hydrolysis activity of RAD-51. PiColorLock™ Gold Colorimetric Assay Kit (Thermo Fisher Scientific) was used to measure ATP hydrolysis activity of RAD-51. RAD-51 (2 µM) was mixed with either RFS-1/RIP-1 (1 or 2 µM) in the presence of ssDNA. The amount of produced inorganic phosphate as a proxy of the ATP hydrolysis activity was analyzed in time.

3. It will be important to show if RFS-1/RIP-1 modulates RAD-51 association and dissociation on

dsDNA by using the same approaches as already described herein, since RAD-51 bound to dsDNA needs to be removed to allow initiation of DNA repair synthesis.

Thank you for this suggestion as it would be indeed interesting to address if RFS-1/RIP-1 also influences RAD-51-dsDNA filaments following strand invasion. We have performed several key experiments described in manuscript using dsDNA in the stop-flow setup instead of ssDNA (see Figure R2 below). We did not observe any clear RFS-1/RIP-1 driven stimulation of RAD-51-dsDNA filament formation either in the presence ATP or ADP (panels **a** and **b** below). We saw potential evidence for RFS-1/RIP-1 destabilization of RAD-51-ssDNA filaments in the presence of ADP (panel **c**), consistent with the reviewer's suggestion. Interestingly however, we observed increased stability of RAD-51-dsDNA filaments in the presence of RFS-1/RIP-1 and ATP (panel **d**), similar to what is seen with RAD-51-ssDNA filaments. We hypothesize that additional factors are likely necessary to help remove RAD-51 and allow initiation of DNA repair synthesis. As this opens up a new project, we would rather omit these results from manuscript. This will also keep the present study focused on the identification and relevance of proofreading activities at the level of HR initiation and prevent dilution of the primary conclusions of the study.

Figure R2. Effect of RFS-1/RIP-1 on formation and dissociation of RAD-51-dsDNA filaments in the presence of ADP (a and c) or ATP (b and d). Average normalized Cy3-43mer fluorescence profiles plotted as a function of time. The arrow indicates the components of the two syringes rapidly mixed at the 0 s time point in a stopped flow instrument. RAD-51 alone (1000 nM in **a** and **b** ; 2000 nM in **c** and **d**) or in the presence of RFS-1/RIP-1 (50 nM) was mixed with 5'-Cy3-dsDNA (43 bp, 15 nM) in the stopped-flow machine in **a** and **b**, or pre-mixed with 5'-Cy3-dsDNA and then mixed with excess of 43mer ssDNA in **c** and **d**. Bar graphs of average Δ Cy3 fluorescence for the RAD-51 alone or in the presence of RFS-1/RIP-1 are plotted (errors: s.d.) (**a**: n=3-4, **b**: n=4, **c**: n=3-5, **d**: n=6-7).

4. Figures 1 and 2. It will be important to include prokaryotic RecA or human RAD51 to establish if the functional interaction between nematode RAD-51 and RFS-1/RIP-1 is species-specific.

We have used human RAD51 and yeast Rad51 proteins to perform the crucial experiments described in Figures 1 and 2 as requested by the reviewer, which are now presented in Supplementary Figure S2 of the revised manuscript. We did not observe any stimulation of filament formation by nematode RFS-1/RIP-1 in the presence of ATP or ADP using human RAD51 (250 nM), suggesting species-specific interactions between nematode RAD-51 and RFS-1/RIP-1 were indeed required (Supplementary Figure S2a-c). In the absence of nucleotides, RFS-1/RIP-1 did not change the final maximal fluorescence during filament formation by human RAD-51 (1000 nM), confirming the ability to prevent RAD-51 binding is also specific to nematode RAD-51 (Supplementary Figure S2d). Yeast Rad51 protein also did not respond to presence of nematode RFS-1/RIP-1 (Supplementary Figure S2e-h). It should be noted that yeast Rad51 does not bind ssDNA in the presence of ADP (250 nM) or absence of nucleotides (1000 nM) (Supplementary Figure S2g-h) however no impact of RFS-1/RIP-1 on yeast Rad51 was observed.

In summary these data confirm that the results presented in Figures 1 and 2 indeed reflect functional species-specific interactions between nematode RAD-51 and RFS-1/RIP-1. We have incorporated these data into Supplementary Figure S2 and pages 6-7 of the revised manuscript.

5. Figure 3a. Other constructs (3'-Cy3 and 5'-Cy3 ssDNA) should be included to establish if RFS-1/RIP-1 destabilizes RAD-51-ssDNA filaments in the presence of ADP.

We had already compared these constructs in Figure 4 of the original manuscript. The destabilization of RAD-51-ssDNA by RFS-1/RIP-1 in the presence of ADP is present on all constructs tested (5'-Cy3, 3'-Cy3 and Int(21)-Cy3), which we discuss in manuscript. We have modified the text in the revised version of the manuscript to clarify this point (page 8: "oligonucleotides labelled at other positions are described below").

6. Supplementary Fig. 2c & d. It is important to include error bars for these experiments to conclude that co-incubation of saturating RAD-51 with RFS-1/RIP-1 does reveal a slight increase in RAD-51 binding to DNA in the presence of both ATP and ADP.

Regarding Supplementary Figure S2c (note, this is now Supplementary Figure S4c), the reviewer's comment is fair, but the main purpose of this figure was to confirm the concentration of RAD-51 required to observe saturated binding to ssDNA on the biosensor. It is just noteworthy that, similar to stopped flow experiments, there was a stimulation of binding at the 250 nM RAD-51 data point, but to prevent confusion, we have removed the relevant sentence from the manuscript. Regarding Supplementary Figure S2d (now Supplementary Figure S4d), as requested by the reviewer we have verified the increased rate of binding is reproducible (n = 3) and corresponding error bars are now included inset in the revised version of the figure.

7. Supplementary Fig. 3. Quantitative analysis and error bars for these experiments are needed.

Thank you for this comment, which now refers to Supplementary Figure S5. We have incorporated the requested quantitative analysis in the manuscript as bar charts corresponding to the dissociation phase of the BLI experiments, demonstrating the results are reproducible (n = 3-4).

8. Page 14. "We previously showed RAD-51-ssDNA filaments...." should be "RAD-51-dsDNA filaments".

Thank you for pointing out this error, which we have corrected in the revised version of the manuscript (page 16).

Reviewer #2 (Remarks to the Author):

Spirek et al report novel attributes of the nematode RAD51 paralog complex RFS1/RIP1 as a molecular switch capable of monitoring and responding to the nucleotide-associated states of RAD51-ssDNA nucleoprotein filaments. Using stop-flow methods and bio-layer interferometry, they have analyzed how RFS1/RIP1 affects RAD51-ssDNA filament dynamics in the presence of ATP and ADP. The main findings are 1) RFS1/RIP1 stimulates RAD51-ssDNA filament formation, 2) RFS1/RIP1 destabilizes ADP-bound RAD51-ssDNA, and thus favors the ATP-bound catalytically active RAD51 filament state, and 3) RFS1 Walker mutants are impaired for this functional attribute. Moreover, the authors provide evidence that the activity of RFS1/RIP1 is responsive to the ATP/ADP ratio.

Based on the results, the authors propose that RFS1/RIP1 fulfills a 'nucleotide proofreading' role in RAD51 filament maintenance. The work has been done well and significantly broadens our understanding of the mechanism of the conserved RAD51 paralogs on RAD51 filament assembly, activity, and maintenance.

There are a few concerns that need to be addressed.

Specific comments

1. Figure 1 shows relative fluorescence changes for 5', 3' or internally Cy3-labelled ssDNA as being different in ATP vs ADP experiments. Do these differences simply reflect experimental variations, or do they indicate differential effects of RAD51 paralog action with respect to fluorophore position?

It is correct that the absolute and relative fluorescence changes in Figure 1 are variable depending on the DNA construct and nucleotide used. For example, the absolute fluorescence changes are stronger for the 5'-Cy3 and 3'-Cy3 constructs in the presence of ATP compared to the Int(21)-Cy3 construct. The protein-induced fluorescence enhancement (PIFE) being monitored is complex and the absolute change depends on the specific local environment of the fluorophore and mode of protein binding which can influence the degree of steric constraint on the dye. For this reason, relative changes introduced by RFS-1/RIP-1 at any given position rather than absolute changes have to be interpreted. We discussed this in Taylor *et al*, Molecular Cell, 2016 in further detail where we first introduced the use of oligonucleotides labelled at different positions to study the polarity of RAD-51-ssDNA filament remodeling and stabilization by RFS-1/RIP-1. In that case, there was a very clear qualitative impact observed in several assays revealing 5'-3' polarity, with some positions in the filament showing very strong regulation by RFS-1/RIP-1 and others only limited.

In Figure 1, while it is possible there are some subtle differential biological effects (perhaps arising due to the influence of RFS-1/RIP-1 on RAD-51 filament growth polarity, reported recently in Belan *et al*, Molecular Cell, 2021), any difference in relative fluorescent changes introduced by RFS-1/RIP-1 at different positions is only clearly seen at 10 nM (for ATP Int(21)-Cy3 in Figure 1d; and for ADP 3'-Cy3 in Figure 1h) and is much less obvious at 50 nM. This is therefore much more subtle than the effects reported in Taylor *et al*, Molecular Cell, 2016 and as such we consider it inconclusive. To make any strong conclusions about potential polarity effects or differential effects with respect of fluorophore position in this assay, we would need to observe a dramatic differential effect at several RFS-1/RIP-1 concentrations at one of the oligonucleotide extreme positions (3'-Cy3 or 5'-Cy3) to be confident a real polar effect was present. This is however ruled out by the present set of experiments in Figure 1. We therefore conclude that RFS-1/RIP-1 stimulates RAD-51 binding throughout the DNA in both the presence of ATP and ADP, but for completeness we have now caveated to text the possibility of some minor polar effects that are beyond the resolution of the assay (pages 4-5: "It should be noted that some minor differences in relative fluorescence changes induced by RFS-1/RIP-1 on the different

positions in the presence of both ATP and ADP (Fig. 1d, h) was also observed, which could possibly reflect polar effects of RFS-1/RIP-1 on RAD-51 filament formation. However these were subtle and not clearly concentration-dependent”).

2. In Fig 2A, how RFS1/RIP1 inhibits RAD51 binding to ssDNA in the absence of nucleotide remains unclear. Is it governed by direct interaction between RAD51 and RFS1? It will be informative to examine if mutants of RAD51 impaired for RFS1 interaction (Taylor et al, Cell, 2015) but proficient in DNA binding in the same analysis.

Thank you for this suggestion. To address this point, we cloned, expressed and attempted purification of two mutants of RAD-51 described previously (V235E and D238A), which demonstrate impaired interaction with RFS-1 in a yeast two-hybrid experiment (Taylor *et al*, Cell, 2015: Figure S6E-F). While we were not able to purify V235E due to its insolubility, we have successfully expressed and purified the D238A mutant, and tested it in the crucial stopped flow experiments. These data are now incorporated in the revised version of the manuscript (page 7) and in new Supplementary Figure S3a-d. In brief, RAD-51 D238A binding to ssDNA is not significantly limited by RFS-1/RIP-1 in the absence of nucleotides (Supplementary Figure S3a,b), confirming the reviewer’s hypothesis that this proofreading activity requires the direct interaction between RAD-51 and RFS-1. For completeness, we also assessed the impact of RFS-1/RIP-1 on RAD-51 D238A binding to ssDNA in the presence of ATP or ADP (Supplementary Figure S3c,d) and found it was also defective. We suggest in the revised text (page 7) that the data are consistent with the reviewer’s suggestion that a direct interaction between RAD-51 and RFS-1/RIP-1 is important to limit RAD-51 binding to DNA in the absence of nucleotide.

3. The data in Fig 2C are inconclusive. DNA binding assays like in Fig S1 or an alternative approach should be employed to support these data.

We are unclear what specific element of Figure 2c is inconclusive. The data in Figure 2c show that dissociation of RAD-51 filament is not affected by RFS-1/RIP-1 complex as the differences in the fluorescence changes are very small and insignificant. In contrast, in Figure 2a we noted a very clear, concentration-dependent impact of RFS-1/RIP-1 on RAD-51 binding to ssDNA in the absence of nucleotide. These results enabled us to conclude only the binding but not dissociation step is influenced by RFS-1/RIP-1 under these conditions. We have now incorporated statistical evaluation in Fig. 2, to support our conclusions. Figure R3 below shows comparison of these data as a bar graph. This confirms RFS-1/RIP-1 mainly influences RAD-51 binding (pre-mix) but not dissociation (post-mix) in the absence of nucleotides. Since stopped flow monitors dissociation in real-time we do not believe EMSA will provide better resolution as it only reveals the status of protein-ssDNA complexes at equilibrium after several minutes of mixing and resolving in agarose gels. We hope the new analysis clarifies this point.

Figure R3. Comparison of Δ Cy3 fluorescence values from stop-flow data shown in Fig. 2b (pre-mix) and 2d (post-mix).

4. In Fig 2E, like WT, the K56A and E138A RFS1 mutants can also inhibit RAD51 binding to ssDNA in the absence of nucleotide. Since these mutants retain interaction with RAD51 (Taylor et al 2015), it will be interesting to examine if RFS1 mutants lacking RAD51 interaction as described in the previous study fail to inhibit Rad51 binding to ssDNA in the absence of nucleotide.

Thanks for this important suggestion, as it correctly points out that RFS-1 mutants K56A and E138A retain some interaction with RAD-51 in yeast two-hybrid experiments. As suggested, we therefore cloned and purified RFS-1/RIP-1 complex incorporating a different mutant of RFS-1 (L134P, Taylor *et al*, Molecular Cell, 2015; the other mutants were insoluble), which is completely abolished for its interaction with RAD-51 in yeast two-hybrid. We repeated the stopped flow experiments with this mutant mixed with wild type RAD-51. These were similar to the experiments mixing the RAD-51 D238A mutant with wild type RFS-1/RIP-1 requested by the reviewer above in point 2. This data is now also presented in Supplementary Figure S3e-h and described on page 7 of the revised manuscript. We indeed observed a profound inability for RFS-1 L134P/RIP-1 in preventing RAD-51 binding to ssDNA in the absence of nucleotide (Supplementary Figure S3e,f). As for RAD-51 D238A, we also extended this analysis to monitor ssDNA binding in the presence of ATP or ADP, and observed RFS-1 L134P/RIP-1 also fails to stimulate wild type RAD-51 binding (Supplementary Figure S3g,h). Combined with the data presented above, this suggests a direct RAD-51/RFS-1 protein-protein interaction is indeed important for these proofreading activities.

5. In Fig 4, the magnitude of destabilization of RAD51-ssDNA filaments (% fluorescence reduction, 4d) by RFS1/RIP1 in the presence of ADP was comparable, but the kinetic changes in 5' - vs 3'- or internal labeled probe were different (Fig 4A-C). Please explain.

Thank you for this comment. We went back to the data and performed a quantitative kinetic analysis by analysing the half-times of the fluorescence change from each trace in Figures 4a-c and have presented this below (Figure R4). However, there is no clear difference in the half-times between the three different ssDNA constructs, similarly to the magnitude of destabilization.

Figure R4. Comparison of observed half-times determined from kinetic traces shown in Figure 4a-c.

6. On page 14, while discussing the potential role of RFS1/RIP1 at stalled replication forks by suggesting priority binding of Rad51 over RPA at very narrow stretches of ssDNA, the model should be discussed within the context of known role of the DNA resection machineries.

This is a great point. We have modified the text to discuss this on page 15: “The RAD51 paralogs thus could have an important role in establishing and maintaining RAD51 in a nucleotide bound state on short naked ssDNA exposed briefly after fork stalling via the mechanisms we describe here, although the DNA resection machinery is also likely to increase ssDNA exposure at stalled forks, making this potential window fairly transitory.”

Reviewer #3 (Remarks to the Author):

In this manuscript, Špírek et al have analyzed how RFS-1/RIP-1 (a nematode RAD51 paralog complex) influences the interaction of RAD-51 with single-stranded DNA (ssDNA) using in vitro “kinetic” assays including rapid-mixing:stopped-flow and Bio-Layer Interferometry. They find that RFS-1/RIP-1 enhances RAD-51 association with ssDNA when experiments are done in the presence of ATP or ADP. However, in absence of nucleotides RFS-1/RIP-1 has the opposite effect, inhibiting RAD-51 association with ssDNA. In terms of effects on RAD-51 dissociation, the authors find that RFS-1/RIP-1 stabilizes RAD-51/ssDNA complexes when ATP is present but promotes dissociation in the presence of ADP. They report that these RFS-1/RIP-1 effects (slow association without nucleotides and faster dissociation of ADP coordinated RAD-51 complexes) occur at sub-stoichiometric ratios relative to RAD-51 monomer concentrations, are not polar, and partially dependent on Walker motifs in RFS-1/RIP-1. The authors propose that RFS-1/RIP-1 has nucleotide proofreading functions.

The idea of “nucleotide proofreading” promoted by the worm RAD51 paralog complex is interesting and will open experimental routes to better understand the molecular functions of RAD51 paralog complexes in vertebrates. The manuscript is well documented and pleasant to read.

Comments and suggestions:

1) The title should say that the findings of this work concern the nematode RAD51 paralog complex.

We have incorporated this suggestion.

2) The first paragraph of the results (p 3) is difficult to understand. Could it be rephrased?

We have made modified the first paragraph for better clarity.

3) Figure 1. What is meant by n= 4-5

This is the number of technical repeats. Thus n=4-5 indicated that each trace is an average of four to five technical repeats (Figures 1b and 1c), and the variability in these technical repeats in turn define the error bars in Figure 1d. The exact number of repeats for any given experimental condition is different, which is why we quote a range in the figure legend.

4) Figure 2a and b should be combined with Figure 1 and should be done at the same RAD-51 concentration

We consider Figures 1 and 2a-b to be making different points and conclusions, which is why we have kept them separate. The former studies the influence of RFS-1/RIP-1 on RAD-51-ssDNA binding in the presence of nucleotides (where stimulation is observed), while the latter looks at the same question but in the absence of nucleotides (where inhibition is observed).

Regarding RAD-51 concentration, the challenge with the reviewer's suggestion is that RAD-51 has intrinsic differences in affinity for ssDNA in the presence and absence of nucleotides, and so to enable a dynamic range for the assay to observe the impact of RFS-1/RIP-1 on RAD-51 binding, different concentrations of RAD-51 are required (250 nM in the former, 1000 nM in the latter). The same issue is also apparent in EMSA experiments – for example in Taylor *et al*, Cell, 2015, we had to use 800 nM RAD-51 to see enough RAD-51 binding to ssDNA in the absence of nucleotide and the impact of RFS-1/RIP-1 (Figure S3E therein), while 200 nM RAD-51 was sufficient in the presence of ATP (Figure 3B therein). As a comparator data point, we have already in fact previously done an experiment very similar to in Figure 1A but at 1000 nM RAD-51 (Taylor *et al*, Cell, 2015: Figure 4D), where we saw a slight increase in RAD-51 binding by RFS-1/RIP-1. Since the experiment was done at saturating RAD-51 conditions the impact was only minor. We discuss this previous data in the first paragraph of the present results section, page 4: "These experiments were always performed in the presence of saturating RAD-51 concentrations (1000 nM) to ensure complete filament formation by RAD-51, thereby increasing the dynamic range for observing filament binding by RFS-1/RIP-1 in the second phase. However, we noticed that RFS-1/RIP-1 also weakly but consistently stimulated the RAD-51 ssDNA-binding first phase under these conditions".

5) Figure 2 legend: "(c)." no period – "as in A" as in a

We have modified the text in the revised version of the manuscript.

6) Sup Fig 2c. RFS-1/RIP-1 does not appear to have major effects on association rates. Could you comment.

This figure (now Supplemental Figure 4c) does not inform on association rates as it only shows how amplitude varies with RAD-51 concentration and not with time. The main point of the figure was to identify the concentration of RAD-51 required for binding saturation. In contrast, Supplementary Figure 4d does show an impact of RFS-1/RIP-1 on association rate, where RFS-1/RIP-1 accelerates

RAD-51 binding. We have now analyzed this quantitatively in response to a request for reviewer 1 which further strengthens this conclusion.

7) Sup Figure 2d; color code is not clear

Thank you for pointing this out. We have modified the color coding of corresponding figure (now Supplementary Figure 4d) to clarify this.

8) BLI experiments of Sup Figure 3 are nice and should be a main figure. Rates should be given.

Thanks for this comment. We have left the figure in Supplementary data for now (Supplementary Figure 5) but are happy to make this a main figure in consultation with the editor. We have also analyzed the rates and incorporated this as bar charts next to the corresponding BLI traces in the revised figure (Supplementary Figure 5).

9) About the quantifications: Could information be retrieved on the ON and OFF rates rather than by reporting the differences between fluorescence end points?

This is a nice idea, but the kinetics of the experimental association and dissociation phases we observe are more complex than they may appear in the traces. Fitting the traces adequately requires complex mathematical equations which probably reflects that RAD-51 filament formation and dissociation are not binary on-off events but contain several phases such as nucleation, isomerization, growth, and dissociation of single RAD-51 protomers and larger oligomers in a single step (Spirek *et al*, 2018, Nucleic Acids Research). As such these formulas are challenging to interpret and therefore, we revert to simpler analyses like differences in fluorescence magnitude and half times between start and end points.

10) Discussion: Strand exchange by RAD51C-XRCC3 and RAD51B-RAD51C. Was this assessed by D-loop assays? If yes it should be noted that D-loop products do not necessarily form through “a strand exchange mechanism” but could be the results of strand annealing activity dependent on the quality of the dsDNA used in the assay.

We have modified the corresponding text in the revised version of the manuscript to reflect the described biochemical activities for these proteins, page 13: “Surprisingly, RAD51C-XRCC3 and RAD51D-XRCC2 human paralog complexes were shown to promote D-loop formation *in vitro*”.

11) A little weakness of the work is the lack of tests in cells for the “proofreading” activity of RFS-1/RIP-1.

The biggest challenge we have here is that RFS-1/RIP-1 has a remarkably complex array of activities within a single small protein complex, which we have systematically analyzed in the present and our prior studies (Taylor *et al*, Cell, 2015 and Molecular Cell, 2016). These include RAD-51 filament stabilization, RAD-51 filament structural remodeling, and the various proofreading activities described in the present study. While we have found mutants defective in these activities, none of these are clean separation of function mutations. This impairs our ability to perform robust *in vivo* studies which we could uniquely and conclusively attribute to the proofreading activities reported here. Another challenge with *in vivo* studies is the inability to distinguish RAD-51 bound to ATP, ADP or unbound to nucleotide co-factors, making it impossible to monitor the differential effects revealed herein.

In Belan *et al*, Molecular Cell, 2021: Figure 6, we recently showed that the RFS-1 mutants K56A and K56R have a clear *in vivo* defect that manifests as increased formation of aberrant RAD-51 foci after

DNA damage. This is attributable in part to their defect in dissociation from RAD-51 filaments (which is required for filament growth), leading to the accumulation of short, dysfunctional RAD-51/RFS-1(K56A/R)/RIP-1 co-complexes in cells. While this goes some way towards addressing the impact of defective *in vivo* RFS-1 function related to its ability to stimulate normal RAD-51 filament formation in the presence of nucleotides, defects associated with RAD-51-ssDNA-ADP destabilization or preventing RAD-51 binding to ssDNA in the absence of nucleotide reported for RFS-1 K56A in the present study could also have contributed to the observed phenotype.

REVIEWERS' COMMENTS

Reviewer #1 (Remarks to the Author):

The authors have strengthened the manuscript significantly with the new data. Only one issue needs to be further clarified.

Fig S3i. The authors concluded that RFS-1/RIP-1 binds to RAD-51 more efficiently in the absence of nucleotides (lane 6) compared to in the presence of ATP (lane 7) or ADP (lane 8). Based on the image of the gel, the amount of RFS-1/RIP-1 is also decreased in the presence of nucleotides (lanes 7 and 8). As such, it is hard to conclude that the physical interaction between RAD-51 and RFS-1/RIP-1 "in solution" is influenced by nucleotide cofactors. Please clarify this issue.

Please double-check that the reference numbers correspond with the context. For example, line 376 should cite reference #39, not #40.

Reviewer #2 (Remarks to the Author):

The authors have taken my points seriously and have done an excellent job addressing them.

The study is very well done and will make a significant contribution toward deciphering the multifaceted role of the conserved Rad51 paralogs in regulating the dynamics of the Rad51-ssDNA presynaptic filament crucial for the execution of DNA break repair by homologous recombination.

Reviewer #3 (Remarks to the Author):

The authors have adequately addressed the comments. The manuscript is acceptable for publication.

RESPONSE TO REVIEWER COMMENTS

We would like to thank reviewers for their appreciation of the effort to address all their comments.

Reviewer #1 (Remarks to the Author):

The authors have strengthened the manuscript significantly with the new data. Only one issue needs to be further clarified.

Fig S3i. The authors concluded that RFS-1/RIP-1 binds to RAD-51 more efficiently in the absence of nucleotides (lane 6) compared to in the presence of ATP (lane 7) or ADP (lane 8). Based on the image of the gel, the amount of RFS-1/RIP-1 is also decreased in the presence of nucleotides (lanes 7 and 8). As such, it is hard to conclude that the physical interaction between RAD-51 and RFS-1/RIP-1 “in solution” is influenced by nucleotide cofactors. Please clarify this issue.

We have quantified the amount of RAD-51 being pull down by RFS-1/RIP-1 and included this in the revised version (Supplementary Fig. S3j). In the presence of ssDNA and ATP, we observed a weakening of this interaction. In contrast, in the absence of DNA the interaction was reduced slightly in the presence of ADP (Supplementary Fig. S3j). Thus, we have accordingly rephrased the conclusion of this data in the revised version.

Please double-check that the reference numbers correspond with the context. For example, line 376 should cite reference #39, not #40.

We thank the reviewer for spotting this and apologize for an error caused by changes in the final formatting step of the revised manuscript. The reference numbering was corrected.

Reviewer #2 (Remarks to the Author):

The authors have taken my points seriously and have done an excellent job addressing them.

The study is very well done and will make a significant contribution toward deciphering the multifaceted role of the conserved Rad51 paralogs in regulating the dynamics of the Rad51-ssDNA presynaptic filament crucial for the execution of DNA break repair by homologous recombination.

Appreciate your view.

Reviewer #3 (Remarks to the Author):

The authors have adequately addressed the comments. The manuscript is acceptable for publication.

Thank you.